# Awake suppression after brief exposure to a familiar stimulus

Ji Won Bang [1,2 ✉] & Dobromir Rahnev [1]

Newly learned information undergoes a process of awake reactivation shortly after the learning offset and we recently demonstrated that this effect can be observed as early as area V1. However, reactivating all experiences can be wasteful and unnecessary, especially for familiar stimuli. Therefore, here we tested whether awake reactivation occurs differentially for new and familiar stimuli. Subjects completed a brief visual task on a stimulus that was either novel or highly familiar due to extensive prior training on it. Replicating our previous results, we found that awake reactivation occurred in V1 for the novel stimulus. On the other hand, brief exposure to the familiar stimulus led to 'awake suppression' such that neural activity patterns immediately after exposure to the familiar stimulus diverged from the patterns associated with that stimulus. Further, awake reactivation was observed selectively in V1, whereas awake suppression had similar strength across areas V1–V3. These results are consistent with the presence of a competition between local awake reactivation and top-down awake suppression, with suppression becoming dominant for familiar stimuli.

---

[1] School of Psychology, Georgia Institute of Technology, Atlanta, GA, USA. [2] Department of Ophthalmology, New York University Grossman School of Medicine, New York, NY, USA. ✉email: JiWon.Bang@nyulangone.org

The human brain has a remarkable ability to learn. It is now understood that this ability depends on post-learning neural processes that have a central role in the consolidation of newly acquired information[1–3]. One of these processes is the later reemergence of the brain activity displayed during learning[4–6]. When this reemergence occurs during periods of wakefulness, it is called "awake reactivation"[4]. Previous studies demonstrated the presence of awake reactivation in the medial temporal lobe[7,8], higher-order association areas[9–13], and even the primary visual cortex V1[14]. This line of research suggests that awake reactivation is likely to occur for many different types of stimuli and across most brain areas.

However, given that most of the brain is involved in the reactivation process, the brain must develop a way to limit the cost of reactivation as much as possible[15]. Indeed, not all new information has to be reactivated. Most of our adult life is spent navigating familiar routes, interacting with familiar people, and completing familiar tasks. Since familiar things are already by definition committed to memory, one possibility is that the brain identifies such experiences and does not reactivate them. This hypothesis is in line with recent research demonstrating the existence of mechanisms that select what information will be reactivated and what information will be ignored. For example, it has been shown that stimuli associated with emotion[13] or reward[16] are more likely to be reactivated. Moreover, a recent study demonstrated that the hippocampus preferentially reactivates new information that has been learned to a lesser degree[17]. This line of research suggests that reactivation is not a homogeneous process uniformly applied to all recently experienced stimuli. Nevertheless, none of these previous studies specifically examined the differences between new and already familiar stimuli and thus it remains unclear whether the brain is indeed able to differentiate between such stimuli when engaging in the process of reactivation.

Here we investigated how stimulus familiarity affects awake reactivation in early visual areas. We first thoroughly familiarized subjects with a particular Gabor patch orientation over at least two days of training. We then briefly exposed subjects to a task based either on the familiar orientation or on a new Gabor patch orientation. To anticipate, we found that immediately after the exposure to the new orientation, there was clear evidence for awake reactivation of that new orientation in V1 but not in V2 or V3. These results replicate our previous findings based on a much longer exposure to a novel orientation[14]. Critically, brief exposure to the already familiar orientation resulted in the opposite effect, a phenomenon that we call "awake suppression." Specifically, the activity patterns in early visual areas were less likely to be classified as the recently exposed familiar stimulus, with this effect having similar strength across areas V1–V3. In addition, greater learning of the familiar orientation was associated with greater awake suppression. These results demonstrate that awake reactivation is not a uniform process applied to all stimuli but that it preferentially occurs for new and unfamiliar information. In fact, our findings suggest that recently encountered familiar stimuli are likely to be suppressed during subsequent periods of restful wakefulness.

## Results

We investigated whether stimulus familiarity affects awake reactivation. Subjects completed a two-interval forced-choice (2IFC) orientation detection task where they indicated which of two intervals contained a Gabor patch of a particular orientation (Fig. 1a). As in our previous research[14], we used two orientations, 45° and 135°. Each subject was extensively familiarized with one of these orientations (randomly chosen among the two) over the

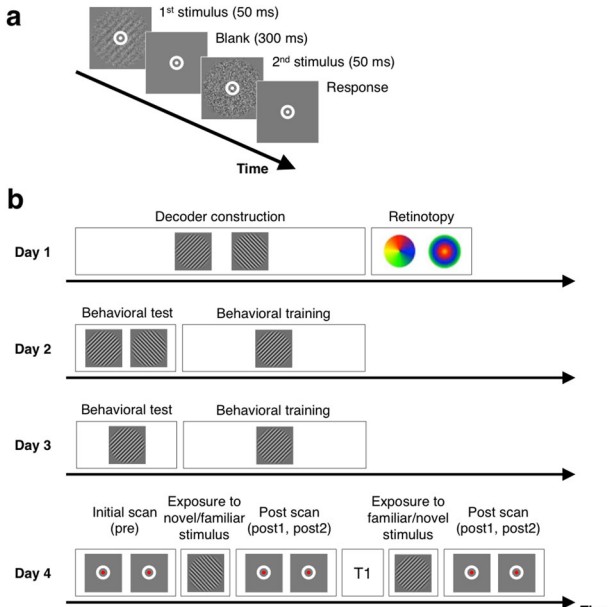

**Fig. 1 Task and experimental procedure. a** Subjects performed a 2IFC orientation detection task. Two stimuli, a target consisting of a Gabor patch embedded in noise and a non-target consisting of pure noise, were presented in random order. Subjects indicated which interval contained the target. **b** The experiment consisted of 4 days. On day 1, we collected decoder construction and retinotopic mapping scans. On days 2 and 3, subjects were familiarized with one Gabor orientation (either 45° and 135°, counterbalanced between subjects) via extensive training on the task using that orientation (several subjects had additional prior training with that orientation; see Methods). Finally, on the critical day 4, we first recorded subjects' brain activity patterns before subjects saw any stimuli (two 5-minute scans; combined into a single "pre" score). We then gave subjects a brief exposure to the task with either the novel or the familiar orientation and then collected the spontaneous brain activity pattern for two more 5-minute scans. After a brief break, during which we collected a T1 scan, we repeated the procedure for the other (familiar or novel) stimulus. The order of exposure to the novel and familiar orientation was counterbalanced across subjects.

course of at least two days of training (days 2–3) with the 2IFC task (Fig. 1b), while the other orientation was not trained on. Finally, on day 4, we briefly exposed subjects to the same 2IFC task once with the familiar and once with the novel orientation (in a counterbalanced order), and analyzed the pattern of brain activity after each exposure.

**Behavioral results**. We first checked whether the extensive training resulted in improved performance on the task. To do so, we compared the behavioral performance on the task on day 2 (before any training had taken place) and day 4 (after the completion of the two days of training). A two-way repeated measures ANOVA with factors time (pre vs. post training) and orientation (familiar vs. novel orientation) showed a significant main effect of time ($F(1,11) = 7.441$, $P = 0.020$, partial $\eta^2 = 0.404$, Supplementary Fig. 1), thus confirming that performance improved after training. Further, the learning amount was not significantly different between the familiar and novel orientations ($T(11) = 0.150$, $P = 0.884$, Hedges' $g = 0.053$, paired $t$ test), suggesting the presence of learning transfer. This generalized learning was observed in our previous study[14], as well as in a lot of prior research[14,18–21].

**Awake reactivation after exposure to a novel stimulus**. We first attempted to replicate our previous results that demonstrated the existence of awake reactivation after a long exposure (about 40 min) to a novel Gabor orientation[14]. To do so, we conducted the same analyses as in our previous paper. Specifically, we constructed a decoder that could distinguish between the familiar and novel orientations for each subject based on the brain activity patterns observed while subjects viewed each Gabor orientation during the decoder construction scan on day 1. We applied this decoder to the brain activity scans performed before (pre: two 5-min scans combined for analysis) and after a brief exposure to a task with the novel orientation (post1 and post2: consecutive 5-min scans). As before, we focused on early visual areas including V1, V2, and V3 because these regions are known to show plasticity after visual training[3,22,23].

Awake reactivation after exposure to the novel orientation would manifest itself in brain activity patterns appearing more similar to the recently exposed novel orientation than the familiar orientation. To test for this effect, we calculated the probability that the decoder would classify the brain activity patterns before and after exposure to the novel orientation as having been produced by either the novel or the familiar orientation. We then conducted a two-way repeated measures ANOVA with factors time (pre vs. post1 vs. post2) and region (V1, V2, V3) to the decoder's classification. As in our previous work[14], the results revealed a significant interaction between time and region ($F_{(4,44)} = 3.094$, $P = 0.025$, partial $\eta^2 = 0.220$; Fig. 2), indicating that the brain activity patterns changed across the three time points and that the pattern of activity was different for the different early visual areas. Further post-hoc tests showed a significant main effect of time in V1 ($F_{(2,10)} = 17.917$, $P < 0.001$, partial $\eta^2 = 0.782$) but not in V2 ($F_{(2,10)} = 0.220$, $P = 0.806$, partial $\eta^2 = 0.042$) or V3 ($F_{(2,10)} = 3.076$, $P = 0.091$, partial $\eta^2 = 0.381$), suggesting that the pattern of activity changed most substantially in V1. We note that Fig. 2 shows that two subjects had lower classification probability in V2 for post1 than the rest of the group, and removing these subjects would make the effects in

V2 similar to V1. Nevertheless, Dixon's test for outliers did not flag these subjects as outliers in the post1 classification in V2 ($P = 0.278$, $Q_{exp} = 0.004$, $Q_{crit} = 0.546$ for the most outlying data) and therefore we have not excluded them in the analyses above.

Given the significant main effect of time in V1, we examined in more detail how the decoder classification changed in V1. We found that neural patterns were classified as the novel orientation significantly more often than chance shortly after the exposure to the novel orientation (post1: $T(11) = 3.452$, $P = 0.005$, Cohen's $d = 0.995$, one-sample $t$ test), but not before (pre: $T(11) = -1.581$, $P = 0.142$, Cohen's $d = 0.456$, one-sample $t$ test) or 5–10 min after the exposure (post2: $T(11) = -1.091$, $P = 0.299$, Cohen's $d = 0.315$, one-sample $t$ test). Moreover, the probability that neural patterns are classified as the novel orientation increased significantly immediately after exposure compared to before exposure (pre vs. post1; $T(11) = -3.209$, $P = 0.008$, Cohen's $d = 0.925$, two-sided paired $t$ test) but this effect disappeared for the second post-exposure scan (pre vs. post2; $T(11) = -0.214$, $P = 0.834$, Cohen's $d = 0.061$, two-sided paired $t$ test). This pattern of results produced a significant quadratic trend indicating a peak at post1 ($F_{(1,11)} = 31.082$, $P < 0.001$, partial $\eta^2 = 0.739$).

We further examined whether these results depend on whether the novel orientation was presented first or second on day 4. A direct comparison of the decoder's classification probability during post1 showed no effect of exposure order in V1 ($T(10) = -0.911$, $P = 0.384$, Hedges' $g = 0.486$, independent sample $t$ test), V2 ($T(10) = 0.410$, $P = 0.690$, Hedges' $g = 0.218$, independent sample $t$ test), or V3 ($T(10) = -0.958$, $P = 0.361$, Hedges' $g = 0.511$, independent sample $t$ test). Critically, the decoder's classification probability for V1 during post1 was comparable between those subjects who were exposed to the novel orientation first (mean = 0.521, SE = 0.013; $T(5) = 1.624$, $P = 0.165$, Cohen's $d = 0.663$, one-sample $t$ test) and those who were exposed to the novel orientation second (mean = 0.536, SE = 0.011; $T(5) = 3.427$, $P = 0.019$, Cohen's $d = 1.399$, one-sample $t$ test), though the probability was significantly higher than chance level for the latter group only. There was similarly

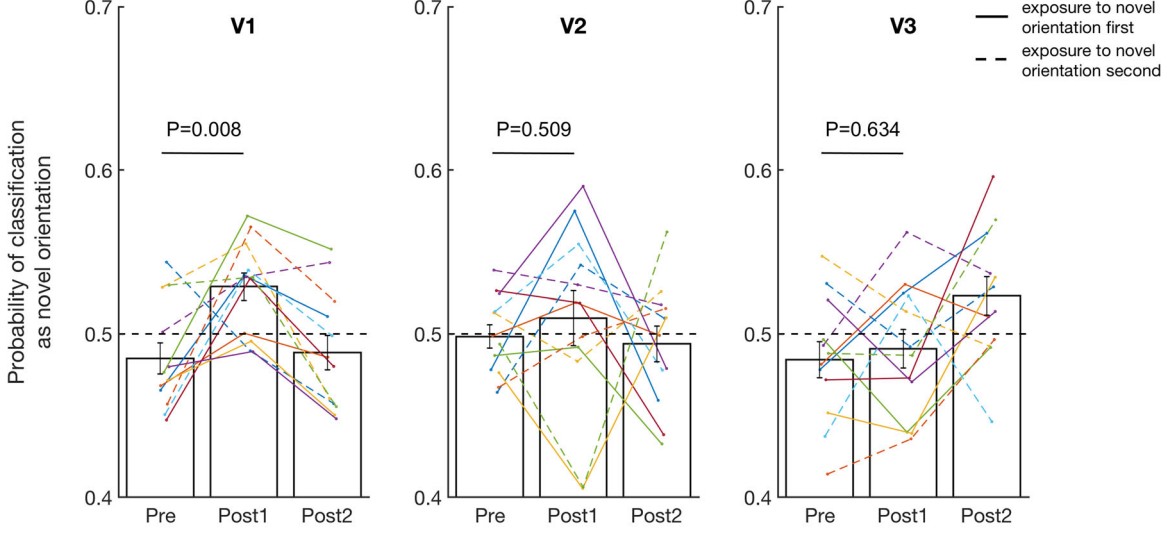

**Fig. 2 Probability that neural patterns are classified as the novel orientation before and after exposure to the novel orientation.** Brain activity was more likely to be classified as the novel orientation in V1 shortly after exposure to the novel orientation. This effect was absent in both V2 and V3. "Pre" refers to two initial scans before subjects saw any stimuli (5 min/scan; combined into a single "pre" baseline). "Post1" and "post2" refer to the first and the second post-task scans immediately after exposure to the novel orientation. The solid and dashed lines represent subjects who were exposed to the novel orientation first and second, respectively. The $P$ values in the figure refer to the results of paired $t$ tests between pre and post1. Error bars indicate s.e.m. $N = 12$.

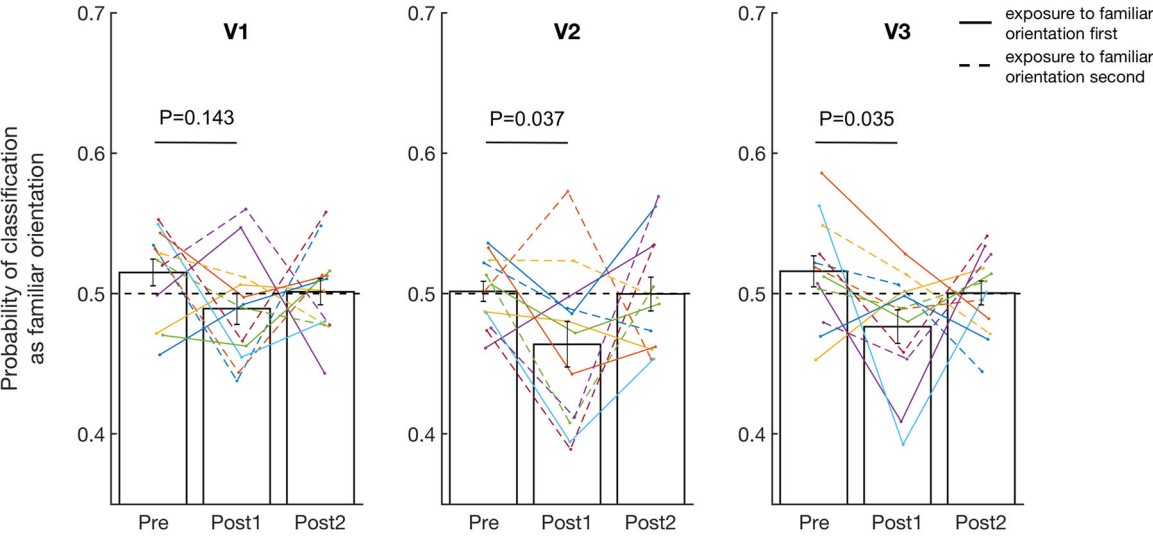

**Fig. 3 Probability that neural patterns are classified as the familiar orientation before and after exposure to the familiar orientation.** Brain activity was less likely to be classified as the familiar orientation across V1–V3 shortly after exposure to the familiar orientation, indicating the presence of awake suppression. "Pre" refers to two initial scans before subjects saw any stimuli (5 min/scan; combined into a single "Pre" baseline). "Post1" and "Post2" refer to the first and the second post-task scans immediately after exposure to the familiar orientation. The solid and dashed lines represent subjects who were exposed to the familiar orientation first and second, respectively. The P values in the figure refer to the results of paired $t$ tests between pre and post1. Error bars indicate s.e.m. $N = 12$.

no effect of exposure order when examining the percentage change of the decoder's classification between before (pre) and after the exposure (post1) (V1: $T(10) = -0.624$, $P = 0.547$, Hedges' $g = 0.332$; V2: $T(10) = -0.367$, $P = 0.721$, Hedges' $g = 0.196$; V3: $T(10) = 0.827$, $P = 0.428$, Hedges' $g = 0.441$; independent sample $t$ tests).

Overall, these results replicate almost exactly our previous findings with a longer (40 min) time of exposure[14] and suggest that awake reactivation occurs in V1 even after much briefer (about 5 min) exposure to a novel orientation. What is more, the decoder was more likely to classify the post1 period as the novel orientation more frequently than pre period for 11 out of the 12 subjects (91.7%, $P = 0.006$, Cohen's $g = 0.417$, two-sided binomial test), suggesting that these effects are extremely consistent across subjects.

**Awake suppression after exposure to a familiar stimulus**. Having confirmed the existence of awake reactivation in V1 after exposure to a novel stimulus, we turned to our main question of whether similar awake reactivation would occur for a familiar stimulus. We repeated the analyses above but now focused on the scans before and after the exposure to the familiar orientation. A two-way repeated measures ANOVA with factors time (pre vs. post1 vs. post2) and region (V1, V2, V3) revealed a significant main effect of time ($F(2,22) = 5.046$, $P = 0.016$, partial $\eta^2 = 0.314$; Fig. 3) but no main effect of region ($F(2,22) = 1.801$, $P = 0.189$, partial $\eta^2 = 0.141$) and no interaction between time and region ($F(4,44) = 0.329$, Huynh–Feldt correction, $P = 0.789$, partial $\eta^2 = 0.029$), indicating that exposure to the familiar orientation changed the classification pattern similarly across the three areas. A further examination of the nature of this change in V1–V3 revealed that the probability of classifying the neural patterns as the familiar orientation decreased significantly immediately after exposure compared to before exposure (pre vs. post1; $P = 0.013$, 95% CI = 0.009−0.060). This effect was the strongest in V2 ($T(11) = 2.377$, $P = 0.037$, Hedges' $g = 0.806$) and V3 ($T(11) =$

2.397, $P = 0.035$, Hedges' $g = 0.918$) and was not significant in V1 ($T(11) = 1.579$, $P = 0.143$, Hedges' $g = 0.672$) though it is important to note that the ANOVA results above showed no evidence for significant differences between the regions and the non-significant effect in V1 may be in part driven by two subjects with high post1 classification probability. Further, the decoder was more likely to classify the post1 period as the familiar orientation less frequently than pre period for 10 out of the 12 subjects in both V2 and V3 (83.3%, $P = 0.039$, Cohen's $g = 0.333$, two-sided binomial test), suggesting that these effects are consistent across subjects.

In other words, we found that exposure to the familiar orientation results in post1 activity being classified as the familiar orientation at a lower than chance level. These results suggest that the familiar orientation was suppressed. However, our classification metric cannot directly distinguish between suppression of the familiar orientation and enhancement of the novel orientation. Nevertheless, enhancement of the novel orientation can only occur after the novel orientation is actually presented. Therefore, to confirm that the results above are indeed due to suppression of the familiar orientation, we examined if the percentage change in the probability of classifying the neural patterns as the familiar orientation from pre to post1 depended on the order of exposure to the familiar and novel orientations. We found no significant difference between subjects who were exposed to the familiar orientation before or after the novel orientation (V1: $T(10) = 1.380$, $P = 0.198$, Hedges' $g = 0.736$; V2: $T(10) = -0.040$, $P = 0.969$, Hedges' $g = 0.021$; V3: $T(10) = -0.325$, $P = 0.752$, Hedges' $g = 0.173$; independent sample $t$ tests). The same results were obtained when V1–V3 were combined into a single region of interest (ROI) with the percentage change being in fact slightly larger for those who were exposed to the familiar orientation first (mean = 7.240) than those who were exposed to the familiar orientation second (mean = 6.790) though the difference was not significant ($T(10) = -0.050$, $P = 0.961$, Hedges' $g = 0.027$, independent sample $t$ test). Nevertheless, this awake suppression effect was not significant in either subgroup individually (both

$P$ values > 0.09; subjects who were exposed to the familiar orientation first: $T(5) = -0.871$, $P = 0.424$, Cohen's $d = 0.356$; subjects who were exposed to the familiar orientation second: $T(5) = -2.030$, $P = 0.098$, Cohen's $d = 1.744$; one-sample $t$ tests) presumably due to the reduced power of these analyses.

Therefore, to further distinguish between awake suppression of the familiar orientation and awake reactivation of the novel orientation, we conducted addition analyses where each of these effects could be examined independently using pattern similarity. We created template patterns for each of the two orientations from the decoder scans and calculated the similarity between the brain activity patterns before and after exposure to the familiar orientation and the template patterns for the familiar orientation. A two-way repeated measures ANOVA with factors time (pre vs. post1 vs. post2) and region (V1, V2, V3) showed that, as in our classification analyses, there was a significant main effect of time ($F(2,22) = 4.618$, $P = 0.021$, partial $\eta^2 = 0.296$; Supplementary Fig. 2) but no main effect of region ($F(2,22) = 0.088$, $P = 0.916$, partial $\eta^2 = 0.008$) or interaction between time and region ($F(4,44) = 0.512$, $P = 0.727$, partial $\eta^2 = 0.044$). A following post-hoc test showed that the similarity for the familiar orientation decreased significantly shortly after the exposure to the familiar stimulus compared to before the exposure (pre vs. post1; $P = 0.018$, 95% CI = 0.001–0.013). An examination of this effect in individual areas showed that the reduction of similarity for the familiar orientation was significant in V3 ($T(11) = 2.825$, $P = 0.017$, Hedges' $g = 0.999$), but not in V1 ($T(11) = 2.092$, $P = 0.060$, Hedges' $g = 0.576$) or V2 ($T(11) = 0.852$, $P = 0.412$, Hedges' $g = 0.264$). This pattern of significance for individual brain areas is slightly different from the pattern of significance in our classification analyses (Fig. 3). However, both overall ANOVAs found no statistical differences between the three areas and therefore the differences in the pattern of statistical significance for individual areas should be interpreted with caution. Importantly, a further two-way repeated measures ANOVA with factors time (pre vs. post1 vs. post2) and region (V1, V2, V3) on the pattern similarity to the novel orientation found no significant effects of time, region, or interaction between time and region (all $P$ values > 0.1; see Supplementary Fig. 3 for statistical values), suggesting that exposure to the familiar stimulus did not result in reactivation of the novel stimulus. Taken together, these pattern similarity results strongly suggest that our binary classification results are due to suppression of the familiar orientation rather than reactivation of the novel orientation.

**Direct comparison of the periods after exposures to novel and familiar stimuli**. The analyses above considered the periods after the exposures to the novel and familiar stimuli separately. For completeness, we also performed a direct comparison of these effects. We conducted a two-way repeated measures ANOVA with factors stimulus type (novel vs. familiar orientation) and region (V1, V2, V3) on the decoder's probability of classifying a stimulus as the recently seen orientation during post1. The results showed a significant effect of stimulus type ($F(1,11) = 5.994$, $P = 0.032$, partial $\eta^2 = 0.353$), indicating that stimulus exposure to the novel and familiar stimuli indeed led to different classification patterns across the different visual areas. The same effect of stimulus type was also observed when instead of post1, we examined the percentage change of the classification probability between pre and post1 ($F(1,11) = 7.821$, $P = 0.017$, partial $\eta^2 = 0.416$). On the other hand, both of these analyses showed no interaction between stimulus type and region (all $P$ values > 0.5; post1 classification: $F(2,22) = 0.679$, $P = 0.518$, partial $\eta^2 = 0.058$; percentage change of the classification between pre and post1:

$F(2,22) = 0.363$, $P = 0.699$, partial $\eta^2 = 0.032$), suggesting that V1, V2, and V3 did not significantly differ in the relative strength of awake reactivation and awake suppression. These results are consistent with the notion that novel and familiar stimuli give rise to different phenomena during the period immediately following stimulus exposure.

**Stronger awake suppression in subjects who exhibited greater learning**. Our results so far strongly suggest that awake suppression occurs in early visual areas after a brief exposure to a familiar stimulus. Given that awake suppression occurs only for familiar stimuli, one may expect that subjects who showed greater awake suppression were the ones who showed greater behavioral improvement during the training phase (days 2–3). To check for such an effect, we computed the performance improvement during training for each subject. We then tested whether subjects with greater vs. lesser performance improvement (based on a median split) exhibit greater awake suppression, defined as the probability of classifying neural activity as the familiar orientation immediately after exposure to the familiar orientation (that is, during post1) within the early visual cortex defined as a single ROI that encompasses areas V1–V3. We found that subjects with greater performance improvement indeed showed larger awake suppression (probability of classification at post1 = 0.437) than subjects with lesser performance improvement (probability of classification at post1 = 0.512; $T(10) = -2.425$, $P = 0.036$, Hedges' $g = 1.292$, independent sample $t$ test; Fig. 4), suggesting that the strength of awake suppression depends on how much learning has previously occurred for the familiar stimulus.

To understand better the nature of the association between awake suppression and the behavioral improvement, we performed two control analyses. First, we confirmed that this association was specific to the learning for the familiar stimulus. Indeed, an equivalent analysis where subjects were split based on the learning on the novel stimulus showed no significant association between performance improvement and the strength of awake suppression ($T(10) = -1.669$, $P = 0.126$, Hedges' $g = 0.890$, independent

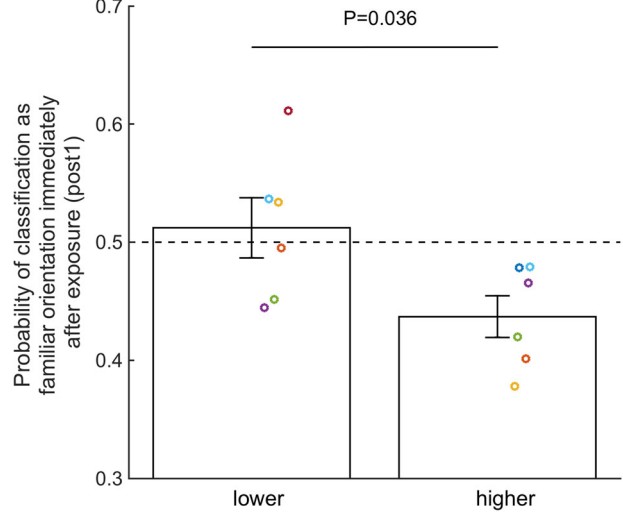

**Fig. 4 Stronger awake suppression in subjects who exhibited greater learning.** Awake suppression was quantified as brain activity after exposure to the familiar orientation that is less likely to be classified as the familiar orientation across V1–V3 during the post1 scan. The awake suppression was significantly stronger for those subjects who showed greater learning on the trained orientation based on a median split. Error bars indicate s.e.m. Dots represent individual data. $N = 12$.

sample $t$ test). A further control analysis showed that there was no significant difference in the strength of awake reactivation during post1 in V1 between subjects with greater vs. lesser performance improvement on the novel orientation ($T(10) = 1.188$, $P = 0.262$, Hedges' $g = 0.636$, independent sample $t$ test). The strength of awake reactivation in V1 would be expected to predict future learning because awake reactivation is thought to be involved in memory consolidation, and indeed we previously found such association[14]. However, the performance improvement metric in the current study indexed the amount of learning before awake reactivation occurred, and therefore the lack of association between the strength of awake reactivation and the performance improvement on the novel orientation is not surprising.

Finally, since different subjects experienced different delays between days 2 and 4 (ranging from three to seven days), we checked whether this variable delay was related to the strength of performance improvement. We divided subjects into those who had a shorter vs. a longer delay between days 2 and 4 using median split and found that those with longer delays showed greater learning ($T(10) = -2.575$, $P = 0.028$, Hedges' $g = 1.372$, independent sample $t$ test). However, the shorter vs. longer delay groups did not differ in the strength of the observed awake suppression ($T(10) = 0.503$, $P = 0.633$, Hedges' $g = 0.268$, unequal variances $t$ test). Therefore, the finding of positive association between awake suppression and learning is unlikely to be driven by the difference in the delay between days 2 and 4.

**No awake reactivation or suppression outside of the early visual areas**. Finally, we explored whether awake reactivation and awake suppression also occur outside of the early visual areas in the context of our stimuli. To do so, we first selected regions where our decoder could reliably distinguish between the two Gabor orientations. We created anatomical ROIs encompassing subareas of the frontal, temporal, and parietal areas, as well as retinotopically defined higher visual areas (see Methods). We performed leave-one-run-out cross-validation (10-fold cross-validation) and found that only V3A, V4v, superior parietal cortex, middle temporal cortex, and fusiform cortex contained decodable information about the stimulus identity. Analyzing each of these regions with one-way repeated measures ANOVAs with a factor time (pre vs. post1 vs. post2) showed no significant main effect of time in any of these ROIs (all $P$ values > 0.1; see Supplementary Figs. 4 and 5 for statistical values). Furthermore, none of these ROIs exhibited significant changes between the pre and post1 periods (all $P$ values > 0.1; see Supplementary Figs. 4 and 5 for statistical values). These results strongly suggest that awake reactivation and awake suppression after exposure to a simple visual stimulus are confined to early visual areas.

## Discussion

We examined whether awake reactivation occurs similarly for novel and familiar stimuli in early visual areas. The results replicated our previous work[14] showing the presence of awake reactivation in V1 after exposure to a novel stimulus. Critically, exposure to a familiar stimulus led to the opposite pattern of results, a phenomenon we call "awake suppression." Further, greater performance improvement on the familiar stimulus was associated with greater awake suppression. These results provide clear evidence that different post-task neural processes occur depending on whether recent experiences were novel or familiar.

Many recent studies have shown that newly encoded information is reactivated during the following rest period[7–14]. This awake reactivation has been proposed to play a key role in consolidating memories[4]. However, reactivating all recent experiences is incredibly expensive and it is likely that complementary

mechanisms determine which experiences should and should not be reactivated. In line with this idea, several previous studies have shown that information paired with reward[16] or shock[13] is preferentially reactivated during following rest. Furthermore, even in the absence of an external factors such as rewards and shocks, the brain appears to be able to select what information to reactivate by, for example, prioritizing weakly learned information[17]. These studies demonstrate that awake reactivation is not a uniform, one-size-fits-all process applied to all recent experiences but that additional mechanisms are employed to modulate when and how reactivation occurs. Nevertheless, this previous research is still consistent with the notion that all recent experiences are reactivated and it is only the strength of this reactivation that is modulated.

The current study is thus the first to demonstrate that recent experiences of familiar stimuli may in fact be suppressed during subsequent offline periods. A related phenomenon of pattern suppression has previously been observed in hippocampus for competing memories during the retrieval of target memories[24] and in early visual cortex for future-irrelevant features during a working memory task[25] but suppression has not previously been demonstrated for post-task periods. Similarly, a recent study investigated the post-encoding period after a task that involved famous vs. non-famous faces (similar to our familiar vs. novel orientation)[26]. The study found that famous and non-famous faces resulted in different patterns of resting state functional connectivity during the post-task period but it is unclear how the altered functional connectivity relates to awake suppression. Thus, our results showing the presence of awake suppression for familiar stimuli reveal a previously unknown phenomenon that is likely to have a critical role in the consolidation of recent experiences into memory.

An alternative interpretation of our finding of awake suppression after exposure to the familiar orientation is that it is instead due to a reactivation of the novel stimulus rather than a suppression of the familiar one. Indeed, both of these effects would have the same effect on our classification results and thus cannot be distinguished directly. Nevertheless, this alternative interpretation is contradicted by several factors. First, the awake reactivation following the exposure to the novel stimulus was short-lived and did not extend into the second post-exposure scan (post2). Therefore, awake reactivation related to the novel orientation is unlikely to appear during the period after exposure to the familiar orientation. Second, half of the subjects were exposed to the familiar orientation first making it unlikely that these subjects would be reactivating a yet-unseen stimulus. Despite this, awake suppression was equally strong in both groups of subjects (though the suppression effect was not significant in either group in isolation). Finally, our pattern similarity analyses confirmed that brain activity after exposure to the familiar orientation was less similar to pattern elicited by the familiar stimulus (consistent with suppression of the familiar stimulus) but not more or less similar to the pattern elicited by the novel stimulus (consistent with a lack of reactivation of the novel stimulus). Thus, there is strong converging evidence that exposure to a familiar orientation results in awake suppression of that orientation in early visual cortex.

An important question that emerges from our results concerns the mechanisms of awake reactivation and awake suppression in early visual areas. One exciting possibility is that our findings of awake reactivation could be due to neuronal replay—that is, reactivation of ensemble firing patterns as experienced during learning. Neuronal replay has been extensively studied in the hippocampus[27–30], prefrontal cortex[31], and visual cortex[32] in animals. Importantly, studies involving animals have also examined such replay in novel vs. familiar spatial environments. These

studies consistently observed that the sequence of firing patterns associated with a novel environment was replayed more strongly than those associated with a familiar environment[27,33,34]. Furthermore, the strength of replay after a novel, but not a familiar, environment predicts later reinstatement during re-exposure[35] and the firing sequences representing a novel environment are more precisely coordinated than those representing a familiar environment[36]. However, none of these studies in animals found suppression of ensemble firing patterns after the exposure to a familiar environment. Therefore, the exact mechanisms at the neuronal level that underlie the awake suppression observed in the current study remain to be elucidated.

Another important question relates to whether awake reactivation and suppression are bottom-up or top-down processes. We previously argued that awake reactivation of specific stimulus orientations is likely a local V1 phenomenon[14]. Indeed, top-down processes, such as visual imagery[37], attention[38], and working memory[39] do not selectively modulate V1 but instead also have a strong influence on higher areas such as V2 and V3. Our current results showing awake reactivation in V1 but not in V2 or V3 provide further evidence for the notion that awake reactivation in our study is likely local to V1. However, awake suppression had similar strength across areas V1–V3 with no significant difference between these regions. This pattern of results is consistent with the interpretation that awake reactivation is a local phenomenon, whereas awake suppression is a top-down process. We propose that there could be continuous competition between local awake reactivation and top-down awake suppression with awake reactivation prevailing for novel stimuli and awake suppression becoming dominant for familiar stimuli. Nevertheless, these possibilities are currently speculative as we did not obtain independent evidence to confirm whether a given effect is local or top-down; further research could address this question by examining, for example, whether awake reactivation and suppression appear in different cortical layers[40,41].

Our results suggest that qualitatively different processes take place for familiar vs. novel stimuli. It is therefore important to clarify what makes a stimulus familiar or novel. Critically, the distinction between the two should not be simply based on time of exposure. For example, your new neighbor who you talked to for five minutes every day for the last seven days can safely be categorized as "familiar," while another new neighbor that you just met and had a 35-minute conversation with can be categorized as "novel" even though you interacted with both for the same total amount of time. This example demonstrates that the critical dimension that distinguishes familiarity from novelty—and thus should govern the presence of awake suppression and awake reactivation—is how firmly the stimulus has already been established in memory with "familiar" stimuli typically requiring exposure over multiple days that would allow strong memory consolidation to take place.

Both awake suppression and awake reactivation in our study were short-lived, only appearing in the first post-exposure period (post1) but not in the second (post2). However, our previous study[14] using a similar paradigm but a much longer exposure period (16 blocks, compared to three blocks in the current study) found that awake reactivation extended into the post2 period. Therefore, it is possible that the duration of the reactivation and suppression depends on the duration of the visual exposure. This possibility is supported by previous research in rodents, which demonstrated that the duration of replay activity depends on the number of stimulus repetitions such that 50 stimulus repetitions led to about 3 min of replay activity, while 125 stimulus repetitions led to about 14 min of replay activity[42]. Relatedly, it is likely that the duration of awake reactivation depends on the typical time scale on which a brain region operates. Previous work has

shown that primary sensory areas operate on a short time scale, while association cortex and the hippocampus operate on much longer time scales[43]. Thus, awake reactivation and suppression are likely to extend further in time in these downstream brain areas.

An important question for future research is whether similar awake suppression occurs in other brain areas involved in awake reactivation such as the hippocampus[7,8] and higher-level association cortex[9–13]. Previous studies have already shown that suppression processes could be studied using fMRI[24,25], suggesting that determining whether awake suppression exists beyond early visual areas is a tractable question. Establishing where in the brain and under what conditions awake suppression occurs would shed additional light on the mechanisms of this phenomenon.

Finally, it should be noted that the finding of both awake reactivation and awake suppression in the same part of the brain rules out a number of trivial explanations for these effects. For example, it is possible that awake reactivation is simply due to subjects engaging in conscious rehearsal of the recently seen information or to certain types of priming effects. However, such effects can explain awake reactivation but not awake suppression. Further, awake suppression occurred even when exposure to the familiar stimulus was given first on day 4, thus arguing against the possibility that subjects were consciously engaging in rehearsal of the novel stimulus (since they had not been exposed to it yet on that day). Thus, the demonstration of awake reactivation and awake suppression in tandem can strengthen the evidence for each of them in isolation.

## Methods

**Participants.** Twelve subjects (19–27 years old, five females) participated in this study. All subjects had normal or corrected-to-normal vision and did not have any history of neurological disorders. All subjects were screened for MRI safety and provided written informed consent. The study was approved by Institutional Review Board of Georgia Institute of Technology. The sample size of 12 can be seen as small in the context of most fMRI studies but is in line with previous studies on visual perceptual learning that involve multi-day designs[11,14,44–51]. Indeed, visual perceptual learning is thought to produce much smaller inter-subject variability than most cognitive tasks making results robust even at relatively small sample sizes, as evidenced by the fact that the current 12-subject study replicated exactly the findings of our previous 12-subject study[14].

**Task.** Subjects performed a 2IFC orientation detection task. Each trial began with a 500-ms fixation period, followed by the two intervals containing the target and non-target stimuli in a random order (50 ms each). The two stimulus intervals were separated by a 300-ms blank period (Fig. 1a). The task was to indicate which of the two intervals contained the target. The target was a Gabor patch of a particular orientation (circular diameter = 10°, contrast = 100%, spatial frequency = 1 cycle/degree, Gaussian filter = 2.5°, random spatial phase). The Gabor patch was masked by noise generated from a sinusoidal luminance. The stimulus intensity was controlled by the ratio of noise pixels. For example, a stimulus with X% S/N ratio contained noise in 100-X% of the pixels of the Gabor patch. The other non-target stimulus consisted of noise only (0% S/N ratio). Subjects reported which interval contained the target by pressing a button on a keypad under no time restriction. No feedback was given after the response.

**Procedures.** The study consisted of 4 separate days (Fig. 1b). On day 1, we collected decoder construction, retinotopy, and anatomical MRI scans. On day 2, subjects were given behavioral tests on the two possible orientations (45° and 135°), followed by extensive training on one of these orientations (which was chosen randomly). Day 3 consisted of the behavioral test and training on the same, previously chosen orientation. Finally, day 4 started with a pre-task scan consisting of two 5-minute scans, which were combined for analyses. We then gave subjects exposure to either untrained (novel) or trained (familiar) orientation in the form of a behavioral test with that orientation, which was immediately followed by two 5-minute post-task scans (post1 and post2). After a short break, during which we collected a T1 anatomical scan, we gave subjects the same exposure but to the other (familiar or novel) orientation and collected the same two 5-minute post-task scans. The order of exposure was pseudo randomized so that exactly half of the subjects were exposed to the novel orientation first. Days 1–2 could be separated by multiple days, whereas days 2–4 were constrained to be performed within a week. The average interval between days 2 and 3 was 3.7 days and that between days 3

and 4 was 2.3 days. Testing during days 1 and 4 was conducted inside an MRI scanner, while testing during days 2 and 3 was performed in a mock scanner.

The behavioral tests on day 2 were performed for both the 45° and 135° orientations with the order of the two orientations determined randomly. Subjects performed three blocks with one orientation, followed by three blocks with the other orientation. The behavioral test on day 3 was performed for only the trained orientation. The purpose of the behavioral test was to measure each individual's threshold S/N ratio for each orientation. The threshold S/N ratio per block was calculated as the geometric mean of the last six reversals as in our previous work[14,52–55]. The behavioral training was performed for one randomly chosen orientation among 45° and 135°. The purpose of the behavioral training was to thoroughly familiarize subjects with just one of the two orientations. The behavioral training on each day consisted of ten blocks. For both the behavioral tests and trainings, each block began with 25% S/N ratio and the following S/N ratios were determined by a 2-down 1-up staircase procedure. Each block terminated after ten reversals. On average, a block consisted of 30–40 trials and lasted 1–2 min.

It should be noted that six subjects had previously participated in a related experiment from our lab[14]. Because our previous study had identical day 1 scans (decoder construction, retinotopy, and anatomical MRI), we did not collect these scans again and simply used the data we had previously collected for these subjects. Further, as part of our previous study, these six subjects were trained on either the 45° or 135° orientation for one session consisting of 16 blocks lasting ~40 min. In the current study, we trained these subjects on the same orientation that they were previously trained on. The time interval between these six subjects' day 1 scans from our previous study[14] and their new data collection in the current study (days 2–4) varied from 5 to 15 months. The relatively long period could decrease the decoder classification probability but such an effect would only make it harder to find evidence for awake suppression or reactivation. Further, the structure and functional properties of the visual cortex are known to be stable during adulthood in the absence of brain injury or visual deprivation[56,57] and substantial MVPA classification accuracy over periods of many months has been observed even in cases of brain injury[58]. Importantly, these six subjects did not differ from the six new subjects in either the observed behavioral or neural effects (Supplementary Table 1). Indeed, there was no significant difference between the two groups in the behavioral improvement for either the familiar ($T(10) = -0.914$, $P = 0.382$, Hedges' $g = 0.487$, independent sample $t$ test) or novel orientation ($T(10) = 0.329$, $P = 0.749$, Hedges' $g = 0.175$, independent sample $t$ test), as well as in either the strength of awake reactivation in V1 ($T(7.084) = 0.509$, $P = 0.626$, Hedges' $g = 0.269$, unequal variances $t$ test) or awake suppression in early visual cortex ($T(10) = -0.576$, $P = 0.578$, Hedges' $g = 0.307$, independent sample $t$ test).

**Decoder construction.** In order to construct a decoder that can distinguish between the two Gabor orientations, we collected a decoder construction scan during which subjects viewed each Gabor orientation (45° and 135°) and performed a task orthogonal to the presented orientation (frequency detection task). Specifically, subjects saw a series of Gabor patches with a given orientation and indicated whether there was a change in the spatial frequency among the presented Gabor patches. The task had 10 runs (1 run = 300 s) each consisting of 18 trials (1 trial = 16 s) with two fixation periods (each 6 s) at the beginning and end of the run. Each trial consisted of a 12-s stimulus presentation period and a 4-s response period. During the stimulus presentation period, 12 Gabor patches with one identical orientation (45° or 135°) flashed at a rate of 1 Hz. Each Gabor patch was presented for 500 ms and there was 500-ms blank period between Gabor patches. In half of all trials, one of the 12 Gabor patches had a higher spatial frequency than the remaining Gabor patches (that had spatial frequency fixed to 1 cycle/degree). Subjects were asked to press a button during the 4-s response period if they detected a change in the spatial frequency among the 12 Gabor patches. In the beginning of the 12-s stimulus presentation period, the central dot changed its color from white to green. This dot remained green during the entire 12-s stimulus presentation period, and then returned to white when the 4-s response period began.

The difficulty of the task was controlled by the degree of the spatial frequency change. The first spatial frequency change in the first run was fixed to 0.24 cycles/degree and the following changes were adjusted via an adaptive staircase method. The spatial frequency change was decreased or increased by 0.02 for hits and misses, respectively. The spatial frequency was not changed for correct rejection or false alarms. The spatial frequency in the following run started from the spatial frequency reached at the end of the previous run. The order of presentation of the two orientations across the 18 trials per run was pseudorandom such that half of the trials had 45° orientation and the other half had 135° orientation.

**Pre-task and post-task scans.** The pre-task scan was collected on day 4 before any task was given. The post-task scan was performed twice, once after the exposures to the familiar and novel orientations. We did not have separate pre-task scans for each exposure because of the possibility that the pre-task scan for the second exposure could be influenced by the first exposure. The pre- and post-task scans consisted of two 5-min scans each. In the analysis, we combined the two 5-min scans of the pre-task scan because we did not expect a change during these

two 5-min scans. On the other hand, we analyzed the two 5-min scans of the post-task scans separately (post1, post2) to examine the time course of awake reactivation and suppression.

During the pre- and post-task scans, subjects performed a fixation task as in our previous work[14]. We used a visual task (as opposed to a task from a different sensory modality) in order to discourage subjects from consciously rehearsing a certain orientation. Subjects were asked to maintain their fixation at the central dot (size and location of the dot was identical with the one used in the 2IFC task) and press a button when they detected a color change of the dot. The central dot's color changed from white to faint pink ([R, G, B] = [255, 255 − x, 255 − x]) for 1.5 s and then returned to white. Subjects had to respond within this 1.5-s interval. Consecutive color changes occurred with an inter-stimulus interval (ISI) ranging between 0.5 and 0.8 s (mean interval = 0.67 s). On average, 138 color change events occurred in each 5-minute scan (one every 2.17 s). The difficulty of the task was controlled by a staircase. The color change x was set to 40 in the beginning and then adjusted via a 2-down 1-up staircase procedure with a step size of 2.

**MRI data acquisition.** MRI data were collected with a Siemens 3 T Trio MR scanner using a 12-channel head coil. Anatomical images were obtained using a T1-weighted MPRAGE sequence (256 slices, voxel size = $1 \times 1 \times 1$ mm³, TR = 2530 ms, FOV = 256 mm). Functional images were collected using a gradient echo-planar imaging sequence (33 slices, voxel size = $3 \times 3 \times 3.5$ mm³, TR = 2000 ms, TE = 30 ms, flip angle = 79°). The slices covered the whole brain and were parallel to the AC-PC plane.

**Retinotopy and ROI selection.** Using standard retinotopy methods[59,60], we presented a flickering checkerboard along the horizontal and vertical meridians and in the upper and lower visual fields. Using contrast maps between horizontal versus vertical meridians and upper versus lower visual fields, we delineated V1, V2, V3, V3A, and ventral V4. We also obtained 27 anatomical ROIs using Freesurfer's cortical parcellation method. For all of these ROIs, we examined whether we can distinguish between the two Gabor orientations using MVPA within the ROI, and then further tested for the presence of awake reactivation and awake suppression in the selected ROIs.

**fMRI data analysis.** We used Freesurfer software (version 6.0) to analyze the MRI data. First, we used the longitudinal stream in Feesurfer[61] to extract reliable structural images from two different days' scans (days 1 and 4). This method generates an unbiased within-subject template using robust, inverse consistent registration. The functional data were preprocessed by applying motion correction, but not spatial or temporal smoothing. The functional images were registered to the individual structural template that was created by the longitudinal stream. Then we used a gray matter mask to extract BOLD signals from the voxels located within the gray matter.

Using Matlab, we shifted the BOLD signals by 6 s to account for the hemodynamic delay. We removed voxels that had spikes greater than 10 SDs from the mean during any one out of ten runs of the decoder construction scan. We also removed a linear trend in the BOLD time course. The BOLD signals in each run were $z$-scored for each voxel. We averaged the BOLD signals of each voxel across 6 volumes (12 s) corresponding to the data from each trial in the decoder construction scan to create the data sample for decoding. We obtained 90 data samples for each orientation, thus in total we had 180 data samples from the 10 runs of the decoder construction scan. Of note, 6 volumes correspond to the Gabor stimulus presentation time during the decoder construction scan.

We used linear sparse logistic regression for decoding[62]. This method implemented by the sparse logistic regression toolbox (SLR toolbox version 1.2.1alpha; http://www.cns.atr.jp/~oyamashi/SLR_WEB.html) selects the relevant voxels in the ROIs and calculates their weights for classification. We trained the decoder to classify the brain activity patterns in each ROI to either 45° or 135° using the 180 data samples. To verify the robustness of the decoder, we tested its reliability using a 10-fold cross-validation where the decoder was re-trained on nine runs and tested on the remaining run. We observed classification probability significantly above chance for our main areas of interest (V1: $T(11) = 8.587$, $P < 0.001$, Cohen's $d = 2.479$; V2: $T(11) = 10.327$, $P < 0.001$, Cohen's $d = 2.981$; V3: $T(11) = 7.287$, $P < 0.001$, Cohen's $d = 2.103$; combined area of V1–V3: $T(11) = 10.200$, $P < 0.001$, Cohen's $d = 2.944$; uncorrected one-sample $t$ tests), as well as for five additional areas (V3A: $T(11) = 7.154$, $P < 0.001$, Cohen's $d = 2.065$; V4v: $T(11) = 6.047$, $P < 0.001$, Cohen's $d = 1.746$; superior parietal cortex: $T(11) = 3.790$, $P = 0.003$, Cohen's $d = 1.094$; middle temporal cortex: $T(11) = 2.794$, $P = 0.017$, Cohen's $d = 0.806$; fusiform cortex: $T(11) = 2.231$, $P = 0.047$, Cohen's $d = 0.644$; uncorrected one-sample $t$ tests).

Once the decoder was created based on the data from the decoder construction scan, we applied it to the pre- and post-task scans from day 4. We used the same voxels that were included during the decoder construction. We shifted the BOLD signals by 6 s, removed a linear trend, $z$-scored, and averaged every 6 volumes. Then we applied the decoder to each 6-volume period of the pre- and post-task scans. The decoder yielded the probability that each 6-volume period was elicited by the familiar or novel orientation. These probability values for each 6-volume period were provided by the output variable named 'Pte' from SLR toolbox. We

averaged these probability scores for either the familiar or novel orientation (depending on the analysis) within pre, post1, and post 2 scans. These averaged probability scores served as the probability of classification of the decoder.

Finally, we note that we did not apply the decoder to the stimulus exposure runs on day 4 because the equivalent (but longer) exposure runs in our previous study (Bang et al. 2018) produced chance-level classification. There are at least three reasons why the decoder cannot be expected to distinguish the two orientations in the exposure runs. First, the stimuli were at threshold for the majority of these runs, making the signal for classification extremely low. Second, the decoder was trained on Gabor patch presentations that lasted for 12 s but during the exposure runs on day 4 the Gabor patch was presented extremely briefly (only for 50 ms per trial) thus further reducing the signal for decoding. Third, our design leads to masking effects whenever the stimulus is presented in the first interval since in such cases the Gabor patch is followed by a noise stimulus a mere 300 ms later. Such masking occurred on approximately half of the trials (whenever the Gabor patch happened to be presented in the first interval) thus further diminishing the strength of the neural representation. The combination of these three factors vastly reduces the BOLD signal for classification. In fact, awake reactivation and suppression can be expected to produce stronger signal for classification since the visual cortex is likely to be reactivating its memory of a high- rather than low-S/N Gabor patch without the presence of any masking from competing stimuli. The lack of above-chance classification during the stimulus exposure runs also means that the awake reactivation observed is not a mere continuation of signals that were already present during stimulus exposure but constitutes the emergence of new signals presumably related to memory processes.

To conduct the pattern similarity analyses, we constructed template patterns for each of the two orientations based on the average multivoxel patterns elicited by each orientation during the decoder scans. Here, we again used the same voxels that were included during the decoder construction. Then we computed how similar each 6-volume period patterns from the pre- and post-task scans are to the template patterns of each orientation using Pearson correlation. We averaged these correlation values separately for the familiar or novel orientation within the pre, post1, and post2 periods. These averaged correlation values were then used to indicate the strength of the pattern similarity.

**Statistics and Reproducibility**. For all statistical tests, we used two-tailed parametric tests. We confirmed that the assumption of normality was not violated for all behavioral performance and classification measures by applying the Kolmogorov–Smirnov goodness of fit test. To assess the equality of variances, we used Levene's test. When the assumption of equal variances was violated, we used unequal variances tests and reported all such cases. In addition, we used Mauchly's sphericity test for all repeated measures ANOVAs to test the assumption of sphericity. When the sphericity assumption was violated, we used Huynh-Feldt correction. We reported all such violations. We included all participants' data in the analysis and described the statistical tests used for each analysis in the relevant section.

**Performance improvement**. We defined the performance improvement for each individual subject based on the change in threshold S/N ratio from before training (day 2) to after training (day 4). We measured the threshold S/N ratio for each day as the mean threshold S/N ratios across the three blocks. We subtracted the threshold S/N ratio after training (day 4) from that before training (day 2) and then divided it by the threshold S/N ratio before training (day 2) to obtain the performance improvement score. This method for calculating performance improvement is common in the field[49,63–66] and is identical to how we calculated performance improvement in our prior work[14,53,54].

**Apparatus**. We created all visual stimuli in Matlab using Psychophysics Toolbox 3[67,68]. We used an LCD display (1024 × 768 resolution, 60 Hz refresh rate) inside a mock scanner and MRI-compatible LCD projector (1024 × 768 resolution, 60 Hz refresh rate) inside the 3 T scanner to present the visual stimuli.

**Reporting Summary**. Further information on research design is available in the Nature Research Reporting Summary linked to this article.

## Data availability
Data from the decoder classification and the pattern similarity measures are freely available at https://osf.io/kmrwf/ (https://doi.org/10.17605/OSF.IO/KMRWF).

## Code availability
Codes for analyses are freely available at https://osf.io/kmrwf/ (https://doi.org/10.17605/OSF.IO/KMRWF).

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

## Acknowledgements
This work was funded by R01MH119189 and R21MH122825.

## Author contributions
J.W.B. and D.R. conceived the idea, designed the experiments, interpreted the results, wrote the manuscript, and reviewed the final draft. J.W.B. conducted experiments and analyzed the data. D.R. supervised the project.

## Competing interests
The authors declare no competing interests.
