## [Peer Review File · Communications Biology]

Reviewers' comments:

Reviewer #1 (Remarks to the Author):

The manuscript by Bang and colleagues asks the question of how awake memory reactivation varies based on stimulus novelty by comparing reactivation evidence for trained versus untrained orientations in visual cortex. The paper is a nice addition to their prior work which found evidence for reactivation of orientation information in V1 but not in later visual cortices (V2, V3). The authors replicate their prior finding of reactivation in V1 for the novel (untrained) orientation and find evidence for 'suppression' or a decrease in the presence of the trained orientation during post-trained-orientation scans. The experiment appears to be sound and the results push forward demonstrations of awake reactivation of information that is related to perceptual learning in primary sensory cortices, and the result of 'suppression' following the trained stimulus exposure is intriguing (although this requires a bit of follow up in my opinion).

Main points:

1. The result of post-encoding 'suppression', or the reduction in the volumes classified as the familiar orientation from pre to post-exposure rest is novel and intriguing. However, this reduction is a bit unexpected. For example, one may hypothesize that there may be a lower level of reactivation evidence for the trained compared to the novel stimulus, perhaps with either a smaller or non-zero increase in classification for the trained stimulus from before to after exposure, but presumably a reduction was not expected. As the authors are already aware, it is possible that this result is in part related to the specific classification method used. Because the authors use a binary classifier that is trained to distinguish between the novel from the familiar orientation (based on the pre-training scans), if evidence for one these orientations increases, evidence for the other orientation will necessarily decrease. It would thus be helpful to understand whether this 'suppression' evidence is tied to the specific binary classification approach used:

- The authors have already (partially) addressed the notion that the suppression of the trained stimulus may be a result of continued reactivation of the novel stimulus, when the novel stimulus is first and the trained stimulus is second. However, this analysis should go a bit further to more clearly show suppression evidence (pre to post-training change in classifier evidence) in each encoding/counterbalance order. Currently the mean evidence is just shown for the post-training period, which is at similar levels across both orders. But please present the magnitude of the change (pre to post) for both orders, and also show statistics for each of those changes, i.e. is the change reliable in each order (although I realize that this analysis will be reduced in power so it is sub-optimal, but important to see nonetheless). This result is currently strongly interpreted in the Discussion (pgs 18-19).

- Another way to address this issue is to move away from binary classification and use a different (complementary) method for assessing reactivation, to understand whether the result is specific to binary classification. It seems like this design would allow the authors to also use a pattern similarity approach, in which template patterns are created for each of the two orientations from the decoder scans, and each set of volumes during the pre and post scans are correlated with the template patterns. This would allow the authors to see evidence for a decrease in similarity with the trained template pattern by comparing the pre and post scans, or alternatively 'classifying' each scan as more similar to one of the orientations without directly opposing them by training a binary classifier. I believe that including such an alternate approach would help to strengthen this novel finding of suppression by showing its robustness across analysis approaches. This approach has been used in the episodic/long-term memory literature examining reactivation which the authors are already familiar with (Staresina et al., 2013; Schapiro et al., 2018; de Voogd et al., 2016).

2. Regarding prior literature that has compared post-learning consolidation activity between novel and familiar information, the authors may want to consider including:

- Liu, Grady, Moscovitch 2017 Neuroimage: this paper compared novel associative encoding and post-encoding periods to a 'famous' condition, which would potentially be somewhat similar to the novel vs. trained orientation used here (although I realize there are many differences as well).
- There are at least a handful of studies that have compared hippocampal reactivation (many during post-learning awake time periods) based on novelty in rodents (typically comparing novel vs. familiar of well-learned spatial environments). Although I realize that the current study is less directly linked with this literature since it examines perceptual learning, the authors may want to include some mentions of this literature: Giri et al., 2019 J Neurosci; van de Ven et al., 2016, Neuron; McNamara et al., 2014 Nature Neurosci; Cheng & Frank, 2008 Neuron; Foster & Wilson, 2006 Nature

Minor points:

1. In the behavioral results, an improvement in performance is found over the course of training in a non-specific fashion (main effect of time and no interaction between time and trained vs. untrained stimulus). Please also provide the breakdown of this (at least in the Supp. Figure) – is the performance improvement with training reliable (significant) both for the trained stimulus and also the untrained stimulus (as suggested by the ANOVA)? This will help the reader to understand how behavior changes with training in this paradigm.

2. In the Fig. 2 legend (and Fig. 3) it may be helpful to mention that the Post1 and Post2 time periods are the scans IMMEDIATELY after exposure to the novel (or familiar) orientation. It took me a little while to parse this (I wasn't sure if Post1 and Post2 referred to the two post-task scans, i.e. both post-novel and post-trained scans, but eventually realized this referred to each of the 5-minute scans immediately after exposure to the novel/trained orientation). Or this could be avoided if more specific terms are used (e.g. Post-Novel1, Post-Novel2, etc.).

3. On pg. 10 the results for novel orientation reactivation are broken down by exposure order (novel first and trained first), but just the post1 classifier evidence is shown. Please also show this in terms of the pre to post1 change in classification of the novel orientation.

4. Along the lines of the prior comment I would also suggest that the ANOVA which compares reactivation differences across trained and novel stimuli are performed on the pre to post1 changes in classification rather than the post classification itself. The experience-dependent change in classification is the most meaningful index here given that participants do already have training and experience with these stimuli in this particular context (which could theoretically influence the baseline presence of these patterns).

5. The differences in suppression for the familiar orientation based on learning are helpful for understanding the functional significance of this effect. However, I wonder about the specificity of this result given that participants improved on the task non-specifically (for both the trained and untrained orientations). Is the difference in suppression based on learning specific to the improvement on the trained stimulus or is this also present if participants are split based on their improvement on the novel orientation (and perhaps indexes learning more broadly)?

6. I also wonder whether the authors performed a similar split-half analysis as in Fig. 4 for the novel orientation reactivation and change in performance. I believe that this would be analogous to a result from their prior 2018 paper which found greater reactivation in V1 for participants with a greater increase in performance over time (although there are some differences in the nature of the training

across the studies).

7. It is helpful that the authors included analysis of multiple ROIs. However, these other ROIs that are considered are only briefly mentioned in the results (they did not show significant reactivation) without showing the results. It would be great to include the results in a supplemental figure, otherwise it is harder to evaluate these null effects.

8. The measure of learning used is the change in performance from Day 2 to 4 of the study. However, a variable number of days was allowed to occur between these time periods (especially between Days 2 and 3 if I am reading the Methods correctly). Did variability in this delay relate to individual differences in learning? This would not influence the comparison between the trained and novel orientation changes, since this is a within-subject factor, but could influence the split half analysis in Fig. 4.

9. The study includes some participants from a prior study. It is stated in the Methods that there were no differences between the new and old participants. It would be helpful to at least report these results – i.e. that the main results of reactivation were similar across these groups. For example, perhaps the suppression could be stronger for these participants given that they have had even more exposure to the trained stimulus?

10. On lines 522-4, voxels are removed if they have excessive variance “during decoder construction scan”. This is little ambiguous because multiple runs were collected, so presumably voxels were excluded if they demonstrated excessive variance in any one out of tens runs, or was this estimated across all runs simultaneously? Please clarify.

11. I found the sentence on lines 525-6 to be a bit confusing and had to rely on other text to understand the procedure for extracting the relevant patterns from the decoder scans. Please clarify, e.g. that the signal was averaged across 6 volumes corresponding to the data from each trial (the use of every 6 volumes implies that the 6 volumes are not averaged together, which I believe is what is occurring).

12. For the color/change detection task performed during the pre and post scans, it is not stated in the Methods how often the color changes occurred (what is the ISI); please include this information. I also wonder whether reactivation evidence is related temporally (within the post scans) to this change in the visual environment. For example, it is possible that this visual stimulus may evoke a general visual response which could in turn reactivate recently seen visual content (as in the notion of ‘pinging’ latent states in visual working memory, Wolff et al., 2017 Nat Neurosci).

13. Lastly, I wonder what the authors make of the lack of apparent reactivation or suppression in the second post-exposure scan, which seems consistent both for the novel and familiar orientation effects. If I remember correctly reactivation was present during both post-exposure scans in their prior 2018 paper. It is somewhat surprising that the duration is relatively short lived, and at least in the long-term memory literature reactivation evidence does not consistently decline within the time window of 5-10 minutes during post-encoding rest periods. Is there something about the specific design which may contribute to this?

Reviewer #2 and #3 (Remarks to the Author):

The authors have analyzed brain activity in humans after experiencing a novel and a familiar stimulus

using fMRI. They first constructed a “decoder” after subjects observed two different orientations of Gabor patches. Subjects were then familiarized with one of the orientations for two days with an orientation detection training. On the final day, subjects were exposed to both trained (familiar), and untrained (novel) stimuli, and their brain activity was analyzed by the decoder after the exposure. The authors reported that the probability of decoder to classify brain signals as a novel stimulus had increased above the chance level after the exposure. This increase is called “awake reactivation”, which replicated the authors’ previous paper in 2018. The authors also reported that the probability of the decoder to classify brain signals as a familiar stimulus had significantly reduced after the exposure. The finding, called “awake suppression”, is a new result. They showed that subjects with higher behavior performance improvement showed greater suppression.

The findings are potentially valuable for understanding how the brain reacts differently after exposure to novel and familiar stimuli. However, there are major concerns, as listed below, that need to be addressed to solidify the authors’ conclusions.

- 1) Lines 99 and 100, the authors concluded as the learning amount was similar between the familiar and novel orientations by comparing the mean S/N threshold values and a learning transfer had occurred. Did the authors compare the performance improvement between familiar and novel orientations?
- 2) Line 146, the authors concluded that the main effect of time is in V1 but not in V2, regarding the decoder’s probability to classify brain activity as the novel stimulus. However, in Figure 2, it looks like V1 and V2 were similar except for two subjects (out of 12). What is the statistical test result if these two subjects are excluded? The distinction between V1 and V2 may not be as clear as concluded.
- 3) Lines 169 and 170, the authors concluded that exposure order does not affect the decoder’s classification probability. Could authors separate those two groups in Figures 2 and 3 with different line styles, or patterns, etc.? It would be easier for readers to visually confirm the similarities.
- 4) Line 172, the authors concluded that the decoder’s performance was comparable between two groups of exposure order. What does decoder performance mean? Could they describe here how it is different than the decoder’s classification probability? Also, as the scores are close to 0.5, did the authors compare the performance of the decoder to the chance level for two groups of exposure order?
- 5) Line 203, in Figure 3, in V1, there are two data points in Post 1, that seem to affect the significance of the T-test. Again, the distinction between V1 and V2 seems driven by two outliers.
- 6) Line 228, the paragraph discusses how the decoder’s probability differed after exposure to familiar and novel orientation. However, no significant interaction of factors (stimulus type and region) was reported. Since V1 for reactivation and V2 for suppression showed significant results when the analyses were done separately, could the authors discuss further the lack of interaction in this analysis?
- 7) Line 247, the authors discuss the results across areas V1-V3. Could they describe this result in more details? Did they pool the probabilities of all regions into one? Was there a difference in suppression between lower and higher performance among regions? Is the not significant result in Figure 3 for V1 due to pooling the results of low and high-performance groups?
- 8) Since the authors reported learning transfer earlier (line 100), did they do the same median split analysis for novel orientation? Could they report and discuss the differences in the probability of classification in novel orientation depending on the learning performance? If there is no difference,

could they discuss why could be the difference between suppression and reactivation?

9) Line 420; the authors had written that the training days could be separated by multiple days and Days 3 and 4 to be separated by up to four days. Could the authors justify that the behavioral performance was not affected by different intervals in between training and testing?

10) Line 439, the authors mentioned that 6 of the subjects had already participated in their previous experiment and did not collect their Day 1 scan again. The time interval was from 5 to 15 months (line 447). Could they cite a source or justify a separation of 5 to 15 months not affecting the decoder performance?

11) Line 447, the authors had mentioned new and old subjects did not show a difference in their behavioral and neural effects. Could the authors create a figure or table showing the results of statistical tests to help the reader?

12) Line 479, the authors had only one pre-task scan at the beginning of Day 4. Could they justify the reason why they did not have another pre-task scan as a baseline after the anatomical scan?

13) Line 486, the authors described a fixation task to prevent conscious rehearsing of a certain orientation. A concern is how an active fixation task affects brain activity in visual areas. Could the authors explain further why the subjects did a visual task but not, for example, an auditory task?

14) Line 547, the authors need to describe in more details how the probability scores are calculated.

15) Line 573, they mentioned using two-tailed parametric tests. However, with a sample size of 12, especially in t-tests for comparing probabilities, it would be helpful to show evidence of normality.

16) Line 578, they described the performance improvement. However, the definition is a bit concerning. They have subtracted the S/N ratio of Day 4 from Day 2 and then divided it by Day 2's S/N ratio. However, for the same absolute change of the S/N ratio, subjects with higher thresholds on Day 2 will be classified as a low improvement. Could the authors describe further their choice of definition?

Reviewer 1

The manuscript by Bang and colleagues asks the question of how awake memory reactivation varies based on stimulus novelty by comparing reactivation evidence for trained versus untrained orientations in visual cortex. The paper is a nice addition to their prior work which found evidence for reactivation of orientation information in V1 but not in later visual cortices (V2, V3). The authors replicate their prior finding of reactivation in V1 for the novel (untrained) orientation and find evidence for ‘suppression’ or a decrease in the presence of the trained orientation during post-trained-orientation scans. The experiment appears to be sound and the results push forward demonstrations of awake reactivation of information that is related to perceptual learning in primary sensory cortices, and the result of ‘suppression’ following the trained stimulus exposure is intriguing (although this requires a bit of follow up in my opinion).

We thank the reviewer for the constructive comments and thorough review.

1. The result of post-encoding ‘suppression’, or the reduction in the volumes classified as the familiar orientation from pre to post-exposure rest is novel and intriguing. However, this reduction is a bit unexpected. For example, one may hypothesize that there may be a lower level of reactivation evidence for the trained compared to the novel stimulus, perhaps with either a smaller or non-zero increase in classification for the trained stimulus from before to after exposure, but presumably a reduction was not expected. As the authors are already aware, it is possible that this result is in part related to the specific classification method used. Because the authors use a binary classifier that is trained to distinguish between the novel from the familiar orientation (based on the pre-training scans), if evidence for one these orientations increases, evidence for the other orientation will necessarily decrease. It would thus be helpful to understand whether this ‘suppression’ evidence is tied to the specific binary classification approach used:

We agree with the reviewer about the inherent difficulties associated with the binary classification approach that we take here. We have now performed both of the analyses suggested by the reviewer below and find additional support for the notion of suppression of the familiar stimulus in early visual cortex after a brief exposure to that stimulus.

- The authors have already (partially) addressed the notion that the suppression of the trained stimulus may be a result of continued reactivation of the novel stimulus, when the novel stimulus is first and the trained stimulus is second. However, this analysis should go a bit further to more clearly show suppression evidence (pre to post-training change in classifier evidence) in each encoding/counterbalance order. Currently the mean evidence is just shown for the post-training period, which is at similar levels across both orders. But please present the magnitude of the change (pre to post) for both orders, and also show statistics for each of those changes, i.e. is the change reliable in each order (although I realize

that this analysis will be reduced in power so it is sub-optimal, but important to see nonetheless). This result is currently strongly interpreted in the Discussion (pgs 18-19).

We agree with the reviewer that calculating the change from pre to post1 is a more meaningful quantity. We have now replaced the previous analyses of post1 with the percentage change between pre and post1 for both orders. We found very similar results. As before, the percentage change of the decoder performance did not differ between subjects who were exposed to the familiar orientation first vs. second (V1: $T(10)=1.380$, $P=0.198$; V2: $T(10)=-0.040$, $P=0.969$, V3: $T(10)=-0.325$, $P=0.752$). The same results was obtained when V1-V3 were combined into a single ROI: the percentage change was comparable between those who were exposed to the familiar orientation first (mean = 7.240) and those who were exposed to the familiar orientation second (mean = 6.790; $T(10)=-0.050$, $P=0.961$) though the effect was not significant in either subgroup individually (both P 's > .05). As the reviewer pointed out, the lack of significant effect in either group in isolation is likely due to the reduced power that we have in these analyses. Overall, we think that these results provide supportive evidence that the order of presentation doesn't matter but we also appreciate the fact that we don't have direct evidence that awake suppression occurs when the group in which the familiar orientation was presented first is analyzed in isolation (we think that such supportive evidence is provided by the next pattern classification analyses we conducted in response to the reviewer's next comment). Therefore, we have toned down our conclusions in the Discussion when discussing this result and explicitly referred to the non-significant results when each group is analyzed in isolation. We paste the updated sections of the Results and Discussion below:

Results

However, our classification metric cannot directly distinguish between suppression of the familiar orientation and enhancement of the novel orientation. Nevertheless, enhancement of the novel orientation can only occur after the novel orientation is actually presented. Therefore, to confirm that the results above are indeed due to suppression of the familiar orientation, we examined if the percentage change in the probability of classifying the neural patterns as the familiar orientation from pre to post1 depended on the order of exposure to the familiar and novel orientations. We found no significant difference between subjects who were exposed to the familiar orientation before or after the novel orientation (V1: $T(10)=1.380$, $P=0.198$; V2: $T(10)=-0.040$, $P=0.969$, V3: $T(10)=-0.325$, $P=0.752$, independent sample t-tests). The same results was obtained when V1-V3 were combined into a single region of interest (ROI) with the percentage change being in fact slightly larger for those who were exposed to the familiar orientation first (mean = 7.240) than those who were exposed to the familiar orientation second (mean = 6.790) though the difference was not significant ($T(10)=-0.050$, $P=0.961$). Nevertheless, this awake suppression effect was not significant in either subgroup individually (both P 's > .05) presumably due to the reduced power of these analyses.

Discussion

Second, half of the subjects were exposed to the familiar orientation first making it unlikely that these subjects would be reactivating a yet-unseen stimulus. Despite this, awake suppression was equally strong in both groups of subjects (though the suppression effect was not significant in either group in isolation).

- Another way to address this issue is to move away from binary classification and use a different (complementary) method for assessing reactivation, to understand whether the

result is specific to binary classification. It seems like this design would allow the authors to also use a pattern similarity approach, in which template patterns are created for each of the two orientations from the decoder scans, and each set of volumes during the pre and post scans are correlated with the template patterns. This would allow the authors to see evidence for a decrease in similarity with the trained template pattern by comparing the pre and post scans, or alternatively ‘classifying’ each scan as more similar to one of the orientations without directly opposing them by training a binary classifier. I believe that including such an alternate approach would help to strengthen this novel finding of suppression by showing its robustness across analysis approaches. This approach has been used in the episodic/long-term memory literature examining reactivation which the authors are already familiar with (Staresina et al., 2013; Schapiro et al., 2018; de Voogd et al., 2016).

We thank the reviewer for suggesting this analysis that can more directly distinguish between suppression of the familiar orientation from reactivation of the novel orientation. We now conducted this additional analysis by creating template patterns for each of the two orientations from the decoder scans. We then calculated how similar the brain activity patterns before and after exposure to the familiar orientation to the template patterns for the familiar orientation. We conducted a two-way repeated measures ANOVA with factors time (pre vs. post1 vs. post2) and region (V1, V2, V3) to the pattern similarity for the familiar orientation. As with our classification analyses, we observed a significant main effect of time ($F(2,22)=4.618$, $P=0.021$, partial $\eta^2=0.296$) but no main effect of region ($F(2,22)=0.088$, $P=0.916$, partial $\eta^2=0.008$) and no interaction between time and region ($F(4,44)=0.512$, $P=0.727$, partial $\eta^2=0.044$). A following post-hoc test showed that the similarity for the familiar orientation decreased significantly shortly after the exposure to the familiar stimulus compared to before the exposure (pre vs. post1; $P=0.018$, 95% CI=0.001-0.013). Moreover, the equivalent ANOVA analysis of the pattern similarity to the novel orientation found no significant effects of time, region, or interaction between time and region (all P 's > 05). Taken together, these pattern similarity results strongly suggest that our binary classification results are due to suppression of the familiar orientation rather than reactivation of the novel orientation. We now include these new analyses in the Results section, and include figures with the pattern similarity analyses in the supplementary. We paste all of these here:

Results

Therefore, to further distinguish between awake suppression of the familiar orientation and awake reactivation of the novel orientation, we conducted additional analyses where each of these effects could be examined independently using pattern similarity. We created template patterns for each of the two orientations from the decoder scans and calculated the similarity between the brain activity patterns before and after exposure to the familiar orientation and the template patterns for the familiar orientation. A two-way repeated measures ANOVA with factors time (pre vs. post1 vs. post2) and region (V1, V2, V3) showed that, as in our classification analyses, there was a significant main effect of time ($F(2,22)=4.618$, $P=0.021$, partial $\eta^2=0.296$; **Supplementary Figure 2**) but no main effect of region ($F(2,22)=0.088$, $P=0.916$, partial $\eta^2=0.008$) or interaction between time and region ($F(4,44)=0.512$, $P=0.727$, partial $\eta^2=0.044$). A following post-hoc test showed that the similarity for the familiar orientation decreased significantly shortly after the exposure to the familiar stimulus compared to before the exposure (pre vs. post1; $P=0.018$, 95% CI=0.001-0.013). Moreover, the equivalent ANOVA analysis of the pattern similarity to the novel orientation found no significant effects of time, region, or interaction

between time and region (all P 's > 0.05; **Supplementary Figure 3**), suggesting that exposure to the familiar stimulus did not result in reactivation of the novel stimulus. Taken together, these pattern similarity results strongly suggest that our binary classification results are due to suppression of the familiar orientation rather than reactivation of the novel orientation.

Supplementary Figure 2. Pattern similarity for the familiar orientation before and after exposure to the familiar orientation. Brain activity was less similar to the familiar orientation across V1-V3 shortly after exposure to the familiar orientation, consistent with the presence of awake suppression. 'Pre' refers to two initial scans before subjects saw any stimuli (5 min/scan; combined into a single 'Pre' baseline). 'Post1' and 'Post2' refer to the first and the second post-task scans immediately after exposure to the familiar orientation. The P values in the figure refer to the results of paired t-tests between pre and post1. Error bars indicate s.e.m.

Supplementary Figure 3. Pattern similarity for the novel orientation before and after exposure to the familiar orientation. Exposure to the familiar orientation did not induce reactivation of the novel orientation. Indeed, two-way repeated measures ANOVAs with factors time (pre vs. post1 vs. post2) and region (V1, V2, V3) showed no significant main effect or interaction (all P 's > 0.05). 'Pre' refers to two initial scans before subjects saw any stimuli (5 min/scan; combined into a single 'Pre' baseline). 'Post1' and 'Post2' refer to the first and the second post-task scans immediately after exposure to the familiar orientation. Error bars indicate s.e.m.

2. Regarding prior literature that has compared post-learning consolidation activity between novel and familiar information, the authors may want to consider including:

- Liu, Grady, Moscovitch 2017 *Neuroimage*: this paper compared novel associative encoding and post-encoding periods to a 'famous' condition, which would potentially be somewhat similar to the novel vs. trained orientation used here (although I realize there are many differences as well).
- There are at least a handful of studies that have compared hippocampal reactivation (many during post-learning awake time periods) based on novelty in rodents (typically comparing novel vs. familiar of well-learned spatial environments). Although I realize that the current study is less directly linked with this literature since it examines perceptual learning, the authors may want to include some mentions of this literature: Giri et al., 2019 *J Neurosci*; van de Ven et al., 2016, *Neuron*; McNamara et al., 2014 *Nature Neurosci*; Cheng & Frank, 2008 *Neuron*; Foster & Wilson, 2006 *Nature*

We thank the reviewer for pointing out these important papers that are very relevant to the general topic of human reactivation. We hadn't included them initially because Liu et al. (2018) demonstrated altered functional connectivity, rather than the reactivation phenomenon itself, while the other studies (Giri et al., 2019; van de Ven et al., 2016; McNamara et al., 2014; Cheng & Frank, 2008; Foster & Wilson, 2006) demonstrated neuronal replay in animals. However, we agree that these studies deserve to be discussed because they are all relevant to our

manuscript. We now discuss all of these studies in some detail in Discussion and paste the relevant sections below:

The current study is thus the first to demonstrate that recent experiences of familiar stimuli may in fact be suppressed during subsequent offline periods. A related phenomenon of pattern suppression has previously been observed for competing memories during the retrieval of target memories (Wimber et al., 2015) but suppression has not previously been demonstrated for post-task periods. Similarly, a recent study investigated the post-encoding period after a task that involved famous vs. non-famous faces (similar to our familiar vs. novel orientation) (Liu et al., 2018). The study found that famous and non-famous faces resulted in different patterns of resting state functional connectivity during the post-task period but it is unclear how the altered functional connectivity relates to awake suppression. Thus, our results showing the presence of awake suppression for familiar stimuli reveal a previously unknown phenomenon that is likely to have a critical role in the consolidation of recent experiences into memory.

...

An important question that emerges from our results concerns the mechanisms of awake reactivation and awake suppression in early visual areas. One exciting possibility is that our findings of awake reactivation could be due to neuronal replay – that is, reactivation of ensemble firing patterns as experienced during learning. Neuronal replay has been extensively studied in the hippocampus (Foster and Wilson, 2006; Diba and Buzsaki, 2007; Davidson et al., 2009; Carr et al., 2011), prefrontal cortex (Euston et al., 2007) and visual cortex (Ji and Wilson, 2007) in animals. Importantly, studies involving animals have also examined such replay in novel vs. familiar spatial environments. These studies consistently observed that the sequence of firing patterns associated with a novel environment was replayed more strongly than those associated with a familiar environment (Foster and Wilson, 2006; McNamara et al., 2014; Giri et al., 2019). Furthermore, the strength of replay after a novel, but not a familiar, environment predicts later reinstatement during re-exposure (van de Ven et al., 2016) and the firing sequences representing a novel environment are more precisely coordinated than those representing a familiar environment (Cheng and Frank, 2008). However, none of these studies in animals found suppression of ensemble firing patterns after the exposure to a familiar environment. Therefore, the exact mechanisms at the neuronal level that underlie the awake suppression observed in the current study remain to be elucidated.

Minor points:

1. In the behavioral results, an improvement in performance is found over the course of training in a non-specific fashion (main effect of time and no interaction between time and trained vs. untrained stimulus). Please also provide the breakdown of this (at least in the Supp. Figure) – is the performance improvement with training reliable (significant) both for the trained stimulus and also the untrained stimulus (as suggested by the ANOVA)? This will help the reader to understand how behavior changes with training in this paradigm.

The direct comparison revealed that the performance was enhanced after training on both the familiar ($T(11)=2.199$, $P=0.050$) and novel orientations ($T(11)=2.230$, $P=0.048$). As requested by the reviewer, we now added this result to Supplementary Figure 1 and are pasting the figure and its legend below:

Supplementary Figure 1. Threshold S/N for the familiar (trained) and novel (untrained) orientations before and after training. To examine if training made subjects better on the task, we performed a two-way repeated measures ANOVA with factors time (pre vs. post) and orientation (trained vs. untrained orientation). The results showed a significant main effect of time ($F(1,11)=7.441$, $P=0.020$, partial $\eta^2=0.404$), suggesting that training improved subjects' performance. The learning amount did not differ between the familiar (trained) and novel (untrained) orientations (interaction between time and orientation: $F(1,11)=0.336$, $P=0.574$, partial $\eta^2=0.030$) and was significant for each orientation (familiar orientation: $T(11)=2.199$, $P=0.050$; novel orientation: $T(11)=2.230$, $P=0.048$). Error bars indicate s.e.m. Dots indicate individual data.

2. In the Fig. 2 legend (and Fig. 3) it may be helpful to mention that the Post1 and Post2 time periods are the scans IMMEDIATELY after exposure to the novel (or familiar) orientation. It took me a little while to parse this (I wasn't sure if Post1 and Post2 referred to the two post-task scans, i.e. both post-novel and post-trained scans, but eventually realized this referred to each of the 5-minute scans immediately after exposure to the novel/trained orientation). Or this could be avoided if more specific terms are used (e.g. Post-Novel1, Post-Novel2, etc.).

We thank the reviewer for pointing out this ambiguity. We now clarify in both figure legends that Post1 (Post2) refers to the first (second) 5-minute scan immediately after exposure to the novel (familiar) orientation. For the reviewer's convenience, we paste both legends below:

Figure 2. Probability that neural patterns are classified as the novel orientation before and after exposure to the novel orientation. Brain activity was more likely to be classified as the novel orientation in V1 shortly after exposure to the novel orientation. This effect was absent in both V2 and V3. 'Pre' refers to two initial scans before subjects saw any stimuli (5 min/scan; combined into a single 'Pre' baseline). 'Post1' and 'Post2' refer to the first and the second post-task scans immediately after exposure to the novel orientation. The solid and dashed lines represent subjects who were exposed to the novel orientation first and second, respectively. The P values in the figure refer to the results of paired t-tests between pre and post1. Error bars indicate s.e.m.

Figure 3. Probability that neural patterns are classified as the familiar orientation before and after exposure to the familiar orientation. Brain activity was less likely to be classified as the familiar orientation across V1-V3 shortly after exposure to the familiar orientation, indicating the presence of awake suppression. 'Pre' refers to two initial scans before subjects saw any stimuli (5 min/scan; combined into a single 'Pre' baseline). 'Post1' and 'Post2' refer to the first and the second post-task scans immediately after exposure to the familiar orientation. The solid and dashed lines represent subjects who were exposed to the familiar orientation first and second, respectively. The P values in the figure refer to the results of paired t-tests between pre and post1. Error bars indicate s.e.m. N.S., not significant.

3. On pg. 10 the results for novel orientation reactivation are broken down by exposure order (novel first and trained first), but just the post1 classifier evidence is shown. Please also show this in terms of the pre to post1 change in classification of the novel orientation.

We agree with the reviewer that it is also important to show the difference in classification between pre and post1. As suggested by the reviewer, we calculated the percentage change of the classification probability between before (pre) and after the exposure (post1) and compared this index between those who were exposed to the novel orientation first and second. We again found no significant group difference in V1 ($T(10)=-0.624$, $P=0.547$, independent sample t-test), V2 ($T(10)=-0.367$, $P=0.721$, independent sample t-test), or V3 ($T(10)=0.827$, $P=0.428$, independent sample t-test). We now included these results in the manuscript and paste the edited section of the Results below:

We further examined whether these results depend on whether the novel orientation was presented first or second on Day 4. A direct comparison of the decoder's classification probability during post1 showed no effect of exposure order in V1 ($T(10)=-0.911$, $P=0.384$, independent sample t-test), V2 ($T(10)=0.410$, $P=0.690$, independent sample t-test), or V3 ($T(10)=-0.958$, $P=0.361$, independent sample t-test). Critically, the decoder's classification probability for V1 during post1 was comparable between those subjects who were exposed to the novel orientation first (mean = 0.521, SE = 0.013) and those who were exposed to the novel orientation second (mean = 0.536, SE = 0.011). Similar results were obtained when examining the percentage change of the decoder's classification between before (pre) and after the exposure (post1) (V1: $T(10)=-0.624$, $P=0.547$; V2: $T(10)=-0.367$, $P=0.721$; V3: $T(10)=0.827$, $P=0.428$; independent sample t-tests). Overall, these results replicate almost exactly our previous findings with a longer (40 minutes) time of exposure (Bang et al., 2018) and suggest that awake reactivation occurs in V1 even after much briefer (about 5 min) exposure to a novel orientation. What is more, the decoder was more likely to classify the post1 period as the novel orientation more frequently than pre period for 11 out of the 12 subjects (91.7%, $P=.006$, two-sided Binomial test), suggesting that these effects are extremely consistent across subjects.

4. Along the lines of the prior comment I would also suggest that the ANOVA which compares reactivation differences across trained and novel stimuli are performed on the pre to post1 changes in classification rather than the post classification itself. The experience-dependent change in classification is the most meaningful index here given that participants do already have training and experience with these stimuli in this particular context (which could theoretically influence the baseline presence of these patterns).

We agree with the reviewer here too. As requested, we computed the percentage changes in classification between pre and post1. Then we performed a two-way repeated measures ANOVA with factors stimulus type (novel vs. familiar orientation) and region (V1, V2, V3) on the percentage changes in the decoder's probability between before (pre) and after the exposure (post1). The results showed a significant main effect of stimulus type ($F(1,11)=7.821$, $P=0.017$, partial $\eta^2=0.416$), suggesting that stimulus exposure to the novel and familiar stimuli led to different classification patterns across the different visual areas. We now added this information in the manuscript. For the reviewer's convenience, we paste the relevant section below:

In the analyses above, we examined the periods after the exposures to the novel and familiar stimuli separately. For completeness, we also performed a direct comparison of these effects. We conducted a two-way repeated measures ANOVA with factors stimulus type (novel vs. familiar orientation) and region (V1, V2, V3) on the decoder's probability of classifying a stimulus as the recently seen orientation during post1. The results showed a significant effect of stimulus type ($F(1,11)=5.994$, $P=0.032$, partial $\eta^2=0.353$), indicating that stimulus exposure to the novel and familiar stimuli indeed led to different classification patterns across the different visual areas. The same effect of stimulus type was also observed when instead of post1, we examined the percentage change of the classification probability between pre and post1 ($F(1,11)=7.821$, $P=0.017$, partial $\eta^2=0.416$).

5. The differences in suppression for the familiar orientation based on learning are helpful for understanding the functional significance of this effect. However, I wonder about the specificity of this result given that participants improved on the task non-specifically (for both the trained and untrained orientations). Is the difference in suppression based on learning specific to the improvement on the trained stimulus or is this also present if participants are split based on their improvement on the novel orientation (and perhaps indexes learning more broadly)?

We appreciate the reviewer for the creative suggestion. We agree with the reviewer that the suppression effect could be associated with general learning, rather than being specific to the trained orientation. To test this possibility, we split subjects based on the performance improvement on the novel orientation. Then we examined whether the suppression effect (measured by probability of classification as familiar orientation immediately after exposure within V1-V3) is significantly different between subjects with greater vs. lesser performance improvement on the novel orientation. The results indicate that the suppression did not differ between groups ($T(10)=-1.669$, $P=0.126$), suggesting that the suppression was specific to the learning on the trained orientation. We now include this result in the manuscript and paste the relevant section below:

This effect was specific to the learning for the familiar stimulus; indeed, an equivalent analysis where subjects were split based on the learning on the novel stimulus showed no significant association between performance improvement and the strength of awake suppression ($T(10)=-1.669$, $P=0.126$).

6. I also wonder whether the authors performed a similar split-half analysis as in Fig. 4 for the

novel orientation reactivation and change in performance. I believe that this would be analogous to a result from their prior 2018 paper which found greater reactivation in V1 for participants with a greater increase in performance over time (although there are some differences in the nature of the training across the studies).

As requested by the reviewer, we examined whether the performance improvement on the novel orientation is associated with the awake reactivation in V1. For this, we compared the probability of classification as novel orientation in V1 between subjects with greater vs. lesser performance improvement on novel orientation. However, we found that the probability of classification as novel orientation was not significantly different between two groups ($T(10)=1.188$, $P=0.262$). This may seem surprising given that our 2018 paper demonstrated that stronger reactivation is associated with greater performance improvement. However, as the reviewer pointed out, there are several critical differences between the studies that preclude a direct comparison. In our previous study, learning occurred during the in-scanner training (on Day 2) and was assessed using the change in performance on tests on different days (Day 1 to Day 3). Thus, in that study, the strength of awake reactivation could reasonably index how much learning was induced by the in-scanner training. On the other hand, the training in the current study took place *before* the fMRI scanning (training was on Days 2 and 3; scanning was on Day 4). Therefore, the analysis above, which assesses the performance improvement between Days 1 and 4 could not possibly be driven by the amount of awake reactivation, because the awake reactivation was only observed after the in-scanner exposure on Day 4. Therefore, replicating the analysis from our 2018 paper would require that we collected a behavioral test on Day 5 and compared how the behavioral improvement between Days 4 and 5 depends on the awake reactivation on Day 4. Unfortunately, however, we did not collect a behavioral test on Day 5 as the purpose of the current paper was different. Nevertheless, even if we collected such a behavioral test on Day 5, the presence of extensive training on a different orientation (and the subsequent transfer to the novel orientation) would have still complicated the interpretation of the results. We now added this analysis and interpretation to the Results section. For the reviewer's convenience, we paste the relevant section below:

A further control analysis showed that there was no significant difference in the strength of awake reactivation during post1 in V1 between subjects with greater vs. lesser performance improvement on the novel orientation ($T(10)=1.188$, $P=0.262$). The strength of awake reactivation in V1 would be expected to predict future learning because awake reactivation is thought to be involved in memory consolidation, and indeed we previously found such association (Bang et al., 2018a). However, the performance improvement metric in the current study indexed the amount of learning *before* awake reactivation occurred, and therefore the lack of association between the strength of awake reactivation and the performance improvement on the novel orientation is not surprising.

7. It is helpful that the authors included analysis of multiple ROIs. However, these other ROIs that are considered are only briefly mentioned in the results (they did not show significant reactivation) without showing the results. It would be great to include the results in a supplemental figure, otherwise it is harder to evaluate these null effects.

We agree with the reviewer that the brief mention of the other ROIs in our previous paper did not allow for the reader to evaluate these results. We now include plots showing the probability of classification as novel or familiar orientation within V3A, V4v, superior parietal cortex, middle temporal cortex and fusiform cortex as Supplementary Figures 4 and 5. In each of the two figures, we indicated, for each area, the P value obtained from the paired t-test between pre and post1. We paste the two figures below:

Supplementary Figure 4. No awake reactivation after exposure to the novel orientation in areas outside V1-V3. We explored whether awake reactivation occurs outside of the early visual areas in the context of our stimuli. To do so, we created ROIs for V3A, V4v, superior parietal cortex, middle temporal cortex and fusiform cortex, which are the regions found to contain decodable information about the stimulus identity. However, brain activity was not more likely to be classified as the novel orientation shortly after exposure to the novel orientation in any of these areas. Indeed, one-way repeated measures ANOVAs with a factor time (pre vs. post1 vs. post2) showed no significant main effect of time in any of these ROIs (all P values > 0.1). Furthermore, none of these ROIs exhibited significant changes between the pre and post1 periods (all P values > 0.4). ‘Pre’ refers to two initial scans before subjects saw any stimuli (5 min/scan; combined into a single ‘Pre’ baseline). ‘Post1’ and ‘Post2’ refer to the first and the second post-task scans immediately after exposure to the novel orientation. The P values in the figure refer to the results of paired t-tests between pre and post1. Error bars indicate s.e.m.

Supplementary Figure 5. No awake suppression after exposure to the familiar orientation in areas outside V1-V3. We explored whether awake suppression occurs outside of the early visual areas in the context of our stimuli. To do so, we created ROIs for V3A, V4v, superior parietal cortex, middle temporal cortex and fusiform cortex, which are the regions found to contain decodable information about the

stimulus identity. However, brain activity was not any less likely to be classified as the familiar orientation shortly after exposure to the familiar orientation in any of these areas. Indeed, one-way repeated measures ANOVAs with a factor time (pre vs. post1 vs. post2) showed no significant main effect of time in any of these ROIs (all P values > 0.1). Furthermore, none of these ROIs exhibited significant changes between the pre and post1 periods (all P values > 0.1). 'Pre' refers to two initial scans before subjects saw any stimuli (5 min/scan; combined into a single 'Pre' baseline). 'Post1' and 'Post2' refer to the first and the second post-task scans immediately after exposure to the familiar orientation. The P values in the figure refer to the results of paired t-tests between pre and post1. Error bars indicate s.e.m.

8. The measure of learning used is the change in performance from Day 2 to 4 of the study. However, a variable number of days was allowed to occur between these time periods (especially between Days 2 and 3 if I am reading the Methods correctly). Did variability in this delay relate to individual differences in learning? This would not influence the comparison between the trained and novel orientation changes, since this is a within-subject factor, but could influence the split half analysis in Fig. 4.

We thank the reviewer for alerting us about the possible role of the delay in affecting learning. The reviewer is correct that variable number of days passed between every two sessions of the experiment. Specifically, the mean interval between Days 2 and 3 was 3.7 days and that between Days 3 and 4 was 2.3 days. To check whether the performance change from Day 2 to Day 4 was affected by the delay between Days 2 and 4, we divided subjects into those who took shorter vs. longer between Days 2 and 4 using median split. We found that those with longer intervals between Days 2-4 showed greater learning ($T(10)=-2.575$, $P=0.028$), suggesting that the delay indeed affected learning. As the reviewer mentioned, the influence on the delay on learning does not affect our awake suppression and awake reactivation results, as those are within-subject factors. Therefore, we checked whether the analysis in Figure 4 where learning was found to be associated with the strength of awake suppression could be driven by the delay between Days 2 and 4. We repeated the analysis from Figure 4 but split our subjects into those who took shorter vs. longer between Days 2 and 4 using median split (same as in the analysis reported above) and checked whether the two groups differ in the strength of awake suppression. We found that there was no significant difference between the two groups ($T(10)=0.503$, $P=0.633$, unequal variances t-test), suggesting that awake suppression was likely driven by the learning and not by the delay, even though those latter two quantities were related to each other. We have now added these analyses and additional information about the delays between the different days to the manuscript and paste the relevant sections below:

Methods

Days 1-2 could be separated by multiple days, whereas Days 2-4 were constrained to be performed within a week. The average interval between Days 2 and 3 was 3.7 days and that between Days 3 and 4 was 2.3 days.

Results

It should be noted that different subjects experienced different delays between Days 2 and 4 (ranging from three to seven days). We therefore checked whether this variable delay was related to the strength of performance improvement. We divided subjects into those who had a shorter vs. a longer delay between Days 2 and 4 using median split and found that those with longer delays showed greater learning

($T(10)=-2.575$, $P=0.028$). However, the shorter vs. longer delay groups did not differ in the strength of the observed awake suppression ($T(10)=0.503$, $P=0.633$, unequal variances t-test). Therefore, the finding of positive association between awake suppression and learning is unlikely to be driven by the difference in the delay between Days 2 and 4.

9. The study includes some participants from a prior study. It is stated in the Methods that there were no differences between the new and old participants. It would be helpful to at least report these results – i.e. that the main results of reactivation were similar across these groups. For example, perhaps the suppression could be stronger for these participants given that they have had even more exposure to the trained stimulus?

We appreciate the reviewer’s request for further details here. We now added the comparison results of awake reactivation, awake suppression and performance improvement between subjects who previously participated vs. the newly recruited subjects. We also added a supplementary table with the means and SDs for further transparency. We paste the relevant section from the Methods and the supplementary table below:

Importantly, these six subjects did not differ from the six new subjects in either the observed behavioral or neural effects (**Supplementary Table 1**). Indeed, there was no significant difference between the two groups in the behavioral improvement for either the familiar ($T(10)=-0.914$, $P=0.382$) or novel orientation ($T(10)=0.329$, $P=0.749$), as well as in either the strength of awake reactivation in V1 ($T(7.084)=0.509$, $P=0.626$) or awake suppression in early visual cortex ($T(10)=-0.576$, $P=0.578$).

	Revisited subjects	New subjects	P value
Performance improvement for novel orientation	0.124 ± 0.060	0.090 ± 0.087	0.749
Performance improvement for familiar orientation	0.058 ± 0.078	0.179 ± 0.107	0.382
Awake reactivation at post1 in V1	0.533 ± 0.007	0.524 ± 0.016	0.622
Awake suppression at post1 in V1-V3	0.464 ± 0.022	0.486 ± 0.032	0.578

Supplementary Table 1. Behavioral and neural measures for newly recruited and revisited subjects. The two groups did not show any difference in the behavioral and neural effects. The numbers reported for each group are mean ± s.e.m. The P values in the table refer to the results of independent sample t-test between two groups.

10. On lines 522-4, voxels are removed if they have excessive variance “during decoder construction scan”. This is little ambiguous because multiple runs were collected, so presumably voxels were excluded if they demonstrated excessive variance in any one out of tens runs, or was this estimated across all runs simultaneously? Please clarify.

We thank the reviewer for pointing out this ambiguity. We removed voxels that had excessive variance during any one out of ten runs. We now clarified this as below:

We removed voxels that had spikes greater than 10 SDs from the mean during any one out of ten runs of the decoder construction scan.

11. I found the sentence on lines 525-6 to be a bit confusing and had to rely on other text to understand the procedure for extracting the relevant patterns from the decoder scans. Please clarify, e.g. that the signal was averaged across 6 volumes corresponding to the data from each trial (the use of every 6 volumes implies that the 6 volumes are not averaged together, which I believe is what is occurring).

We appreciate the reviewer for pointing out this ambiguity. The reviewer has the correct interpretation and we have now edited this sentence to clarify the intended meaning as below:

We averaged the BOLD signals of each voxel across 6 volumes (12 s) corresponding to the data from each trial in the decoder construction scan to create the data sample for decoding.

12. For the color/change detection task performed during the pre and post scans, it is not stated in the Methods how often the color changes occurred (what is the ISI); please include this information.

We thank the reviewer for alerting us that we did not provide sufficient details about the color change task. The color change occurred very frequently – on average 138 color change events occurred in the 5-minute scan (one every 2.17 seconds on average). We now provide additional details about the timing and frequency of color change occurrence. The relevant section from the Methods is pasted below:

The central dot’s color changed from white to faint pink ([R, G, B] = [255, 255 – x, 255 – x]) for 1.5 s and then returned to white. Subjects had to respond within this 1.5-s interval. After this 1.5-s interval, the following color change occurred after a random interval ranging between 0.5 and 0.8 seconds (mean interval = 0.67 seconds). On average, 138 color change events occurred in each 5-minute scan (one every 2.17 seconds). The difficulty of the task was controlled by a staircase. The color change x was set to 40 in the beginning and then adjusted via a 2-down 1-up staircase procedure with a step size of 2.

I also wonder whether reactivation evidence is related temporally (within the post scans) to this change in the visual environment. For example, it is possible that this visual stimulus may evoke a general visual response which could in turn reactivate recently seen visual content (as in the notion of ‘pinging’ latent states in visual working memory, Wolff et al., 2017 Nat Neurosci).

We agree with the reviewer that this is an exciting possibility. Unfortunately, the current study is not suitable to address this question because the color change occurred very frequently (on average every 2.17 seconds) while the TR was 2 seconds. Therefore, every TR temporally overlaps with at least one color change event, which means that we don’t have any data where a color change didn’t occur. We plan to change this task in future studies by making the color changes a lot less infrequent thus allowing us to examine whether awake reactivation or awake suppression relate in any way to the visual display.

13. Lastly, I wonder what the authors make of the lack of apparent reactivation or suppression in the second post-exposure scan, which seems consistent both for the novel and familiar orientation effects. If I remember correctly reactivation was present during both post-exposure scans in their prior 2018 paper. It is somewhat surprising that the duration is relatively short lived, and at least in the long-term memory literature reactivation evidence does not consistently decline within the time window of 5-10 minutes during post-encoding rest periods. Is there something about the specific design which may contribute to this?

The reviewer is correct that the awake reactivation in our 2018 paper persisted into the post2 scan, unlike in the current study. We suspect that the duration of the reactivation and suppression depends on the duration of the visual stimulus training. If so, this would explain the difference between the duration of reactivation because our 2018 paper used extensive training involving 16 blocks in total, while the current study only included a brief exposure consisting of 3 blocks. This possibility is supported by previous research in rodents, which demonstrated that the duration of replay activity depends on the number of stimulus repetitions (Han et al., 2008). In that study, 50 stimulus repetitions led to about 3 minutes of replay activity, while 125 stimulus repetitions led to about 14 minutes of replay activity. This study is thus consistent with the possibility that the relatively shortened time window of reactivation in the current study may be driven by the smaller number of blocks compared to our previous study.

Another question brought up by the reviewer is why the reactivation observed here is shorter than what is typically reported in the long-term memory literature. While it is difficult to know with certainty, it is likely that the duration of reactivation depends on the typical time scale on which a region operates. Indeed, a lot of work (e.g., from Uri Hasson’s lab) has shown that primary sensory areas operate on a short time scale, while association cortex and the hippocampus operate on much longer time scales (e.g., Hasson et al., 2008). It is possible that this property of the different cortical regions extends to the duration on which reactivation occurs.

We now discuss both of these issues in a new paragraph in the Discussion that we paste below:

Both awake suppression and awake reactivation in our study were short-lived, only appearing in the first post-exposure period (post1) but not in the second (post2). However, our previous study (Bang et al., 2018) using a similar paradigm but a much longer exposure period (16 blocks, compared to 3 blocks in the current study) found that awake reactivation extended into the post2 period. Therefore, it is possible that the duration of the reactivation and suppression depends on the duration of the visual exposure. This possibility is supported by previous research in rodents, which demonstrated that the duration of replay activity depends on the number of stimulus repetitions such that 50 stimulus repetitions led to about 3 minutes of replay activity, while 125 stimulus repetitions led to about 14 minutes of replay activity (Han et al., 2008). Relatedly, it is likely that the duration of awake reactivation depends on the typical time scale on which a brain region operates. Previous work has shown that primary sensory areas operate on a short time scale, while association cortex and the hippocampus operate on much longer time scales (e.g., Hasson et al., 2008). Thus, awake reactivation and suppression are likely to extend further in time in these downstream brain areas.

Reviewer 2

The authors have analyzed brain activity in humans after experiencing a novel and a familiar stimulus using fMRI. They first constructed a “decoder” after subjects observed two different orientations of Gabor patches. Subjects were then familiarized with one of the orientations for two days with an orientation detection training. On the final day, subjects were exposed to both trained (familiar), and untrained (novel) stimuli, and their brain activity was analyzed by the decoder after the exposure. The authors reported that the probability of decoder to classify brain signals as a novel stimulus had increased above the chance level after the exposure. This increase is called “awake reactivation”, which replicated the authors’ previous paper in 2018. The authors also reported that the probability of the decoder to classify brain signals as a familiar stimulus had significantly reduced after the exposure. The finding, called “awake suppression”, is a new result. They showed that subjects with higher behavior performance improvement showed greater suppression.

The findings are potentially valuable for understanding how the brain reacts differently after exposure to novel and familiar stimuli. However, there are major concerns, as listed below, that need to be addressed to solidify the authors’ conclusions.

We thank the reviewer for the constructive comments and thorough review. We have now addressed all of the reviewer’s concerns in detail below.

1) Lines 99 and 100, the authors concluded as the learning amount was similar between the familiar and novel orientations by comparing the mean S/N threshold values and a learning transfer had occurred. Did the authors compare the performance improvement between familiar and novel orientations?

We agree that a direct test of performance improvement between the familiar and novel orientations is more appropriate than the test that we initially reported. Therefore, we now report the comparison between the performance improvement for the familiar and novel orientations. We found that the performance improvement was comparable between the familiar (mean = 0.119, SE = 0.066) and novel orientations (mean=0.107, SE=0.051; $T(11)=0.150$, $P=0.884$, paired t-test), suggesting the presence of learning transfer. We now include this result in the main text.

2) Line 146, the authors concluded that the main effect of time is in V1 but not in V2, regarding the decoder's probability to classify brain activity as the novel stimulus. However, in Figure 2, it looks like V1 and V2 were similar except for two subjects (out of 12). What is the statistical test result if these two subjects are excluded? The distinction between V1 and V2 may not be as clear as concluded.

We thank the reviewer for pointing out the two subjects who showed different patterns in V2 at post1. As requested by the reviewer, we compared the probability of classification as novel orientation between pre and post1 in V2 after excluding these two subjects. The direct comparison revealed close to significant difference ($T(9)=-2.234$, $P=0.052$). However, Grubb's test for outliers did not detect any outliers in the data for V2's post1 ($P>0.3$). Therefore, we do not think that it is appropriate to exclude these two subjects in our analyses a priori. Nevertheless, we now mention these two subjects in the text and note that the results would change if they were to be excluded. We paste the relevant section below:

We note that Figure 2 shows that two subjects had lower classification performance in V2 for post1 than the rest of the group, and removing these subjects would make the effects in V2 similar to V1. Nevertheless, Grubb's test for outliers did not flag these subjects as outliers in the post1 classification in V2 ($P>0.3$) and therefore we have not excluded them in the analyses above.

3) Lines 169 and 170, the authors concluded that exposure order does not affect the decoder's classification probability. Could authors separate those two groups in Figures 2 and 3 with different line styles, or patterns, etc.? It would be easier for readers to visually confirm the similarities.

We thank the reviewer for this suggestion. Now we separated two groups in Figures 2 and 3 by using different line styles. These new figures allow the reader to more easily appreciate that there are no systematic differences between the two groups of subjects. For the reviewer's convenience, we pasted Figures 2 and 3 below:

4) Line 172, the authors concluded that the decoder's performance was comparable between two groups of exposure order. What does decoder performance mean? Could they describe here how it is different than the decoder's classification probability?

We thank the reviewer for pointing out this ambiguity. We used the terms 'decoder performance' and 'decoder's classification probability' interchangeably but recognize that this

is confusing because it seems like the two phrases refer to different quantities. We now changed the phrase 'decoder's performance' to the 'decoder's classification probability' throughout the manuscript. The word 'performance' is now only used in the context of the behavioral results. We think that this has increased the clarity of our manuscript.

Also, as the scores are close to 0.5, did the authors compare the performance of the decoder to the chance level for two groups of exposure order?

As requested by the reviewer, we compared the decoder's classification probability to the chance level (0.5) for each group of exposure order. The decoder's classification probability was significantly above the chance level for those who were exposed to the novel orientation second (mean=0.536, SE=0.011; $T(5)=3.427$, $P=0.019$), but not for those who were exposed to the novel orientation first (mean=0.521, SE=0.013; $T(5)=1.624$, $P=0.165$). However, as stated in the manuscript, the direct comparison between two groups showed no significant difference ($T(10)=-0.911$, $P=0.384$, independent sample t-test). It is therefore unlikely that the fact that only one group has p value below .05 means that the groups differ significantly from each other. We have included this result in the manuscript.

5) Line 203, in Figure 3, in V1, there are two data points in Post 1, that seem to affect the significance of the T-test. Again, the distinction between V1 and V2 seems driven by two outliers.

We thank the reviewer for pointing out two data points which showed different patterns in V1's post1. As suggested, we compared the decoder's classification probability between pre and post1 after excluding these two subjects. A direct comparison showed significant difference between pre and post1 ($T(9)=2.460$, $P=0.036$). However, the Grubb's test for outlier did not detect any outliers in the data for V1's post1 ($P>0.3$). Therefore, we do not think that it is appropriate to exclude these two subjects in our analyses a priori. Nevertheless, we want to clarify that since we found no main effect of region or interaction between time and region, we have not made any claims that awake suppression differs between regions (e.g., we do not think that it is different between V1 and V2). We now specifically note that our results should not be interpreted as showing a meaningful difference between the regions and mention the two subjects noted by the reviewer. We paste the relevant section below:

This effect was the strongest in V2 ($T(11)=2.377$, $P=0.037$) and V3 ($T(11)=2.397$, $P=0.035$) and was not significant in V1 ($T(11)=1.579$, $P=0.143$) though it is important to note that the ANOVA results above showed no evidence for significant differences between the regions and the non-significant effect in V1 may be in part driven by two subjects with high post1 classification probability.

6) Line 228, the paragraph discusses how the decoder's probability differed after exposure to familiar and novel orientation. However, no significant interaction of factors (stimulus type

and region) was reported. Since V1 for reactivation and V2 for suppression showed significant results when the analyses were done separately, could the authors discuss further the lack of interaction in this analysis?

We thank the reviewer for bringing up this important point. As explained above, we found no evidence for a difference between V1, V2, and V3 in awake suppression and don't claim that such difference exists. It is true that $P < .05$ for V2 and V3 and $P > .05$ for V1 but this does not mean that V1 differs from V2 and V3. We have now clarified this in the response above and the main text. On the other hand, when it comes to awake reactivation, we did find a significant interaction between time and region, which also replicates our previous findings (Bang et al., 2018). When comparing the awake reactivation and awake suppression to each other, we don't see any significant effect related to region, which means that there is no significant difference between how awake reactivation and awake suppression manifest in V1, V2, and V3. We have now explicitly clarified this in the paragraph in the Results section. Further, given the reviewer's comments, we have gone over the entire manuscript one more time and made sure to clarify everywhere that we do not claim that V1, V2, and V3 are significantly different in anything other than the strength of awake reactivation. We paste the new sentence in the relevant paragraph here:

On the other hand, both of these analyses showed no interaction between stimulus type and region (all P 's $> .05$), suggesting that V1, V2, and V3 did not significantly differ in the relative strength of awake reactivation and awake suppression.

7) Line 247, the authors discuss the results across areas V1-V3. Could they describe this result in more details? Did they pool the probabilities of all regions into one?

We thank the reviewer for pointing out the ambiguity in the description here. For this analysis, we combined V1, V2, and V3 into a single region of interest (ROI). Then we trained the decoder and applied the trained decoder to the brain activity within this ROI. The decoder yielded the probability of classification for familiar orientation in this ROI covering V1-V3. We now clarified this in the manuscript and we paste the relevant section below:

We then tested whether subjects with greater vs. lesser performance improvement (based on a median split) exhibit greater awake suppression, defined as the probability of classifying neural activity as the familiar orientation immediately after exposure to the familiar orientation (that is, during post1) within the early visual cortex defined as a single ROI that encompasses areas V1-V3.

Was there a difference in suppression between lower and higher performance among regions? Is the not significant result in Figure 3 for V1 due to pooling the results of low and high-performance groups?

Because there was no difference between the regions in awake suppression, we created a single ROI covering all three areas. At the reviewer's request, we additionally examined the

difference in awake suppression between subjects with lesser vs. greater performance improvement within each area and found no significant effects within any individual area (all P 's > .05). This result is consistent with our previous results showing that awake suppression is not driven by any specific area among V1, V2, and V3. Additionally, re-examining the strength of awake suppression in V1, we found that the suppression was comparable between subjects who showed greater learning (mean = 0.489, SE = 0.017) and subjects who showed lesser learning (mean = 0.490, SE=0.016). We have not currently included these results in the paper but we believe that the clarifications already mentioned above that we do not claim that V1 and V2 show a difference in awake suppression already address the reviewer's concern. Nevertheless, if the reviewer thinks that these results help the reader, we would be happy to add them to the manuscript in a further revision.

8) Since the authors reported learning transfer earlier (line 100), did they do the same median split analysis for novel orientation? Could they report and discuss the differences in the probability of classification in novel orientation depending on the learning performance? If there is no difference, could they discuss why could be the difference between suppression and reactivation?

As requested by the reviewer, we examined whether the performance improvement on the novel orientation is associated with the awake reactivation in V1. For this, we compared the probability of classification as novel orientation in V1 between subjects with greater vs. lesser performance improvement on novel orientation. However, we found that the probability of classification as novel orientation was not significantly different between two groups ($T(10)=1.188$, $P=0.262$). This may seem surprising given that we previously demonstrated that stronger reactivation is associated with greater performance improvement (Bang et al., 2018). However, in our previous study, learning occurred during the in-scanner training (on Day 2) and was assessed using the change in performance on tests on different days (Day 1 to Day 3). Thus, in that study, the strength of awake reactivation could reasonably index how much learning was induced by the in-scanner training. On the other hand, the training in the current study took place *before* the fMRI scanning (training was on Days 2 and 3; scanning was on Day 4). Therefore, the analysis above, which assesses the performance improvement between Days 1 and 4 could not possibly be driven by the amount of awake reactivation, because the awake reactivation was only observed after the in-scanner exposure on Day 4. We now added this analysis and interpretation to the Results section. For the reviewer's convenience, we paste the relevant section below:

A further control analysis showed that there was no significant difference in the strength of awake reactivation during post1 in V1 between subjects with greater vs. lesser performance improvement on the novel orientation ($T(10)=1.188$, $P=0.262$). The strength of awake reactivation in V1 would be expected to predict future learning because awake reactivation is thought to be involved in memory consolidation, and indeed we previously found such association (Bang et al., 2018a). However, the performance improvement metric in the current study indexed the amount of learning *before* awake reactivation occurred, and therefore the lack of association between the strength of awake reactivation and the performance improvement on the novel orientation is not surprising.

9) Line 420; the authors had written that the training days could be separated by multiple days and Days 3 and 4 to be separated by up to four days. Could the authors justify that the behavioral performance was not affected by different intervals in between training and testing?

We thank the reviewer for pointing out the possible effects of the intervals between training and test on the behavioral performance. To check whether the performance change from Day 2 to Day 4 was affected by the delay between Days 2 and 4, we divided subjects into those who took shorter vs. longer between Days 2 and 4 using a median split. We found that those with longer intervals between Days 2-4 showed greater learning ($T(10)=-2.575$, $P=0.028$), suggesting that the delay indeed affected learning. This result does not affect in any way our main analyses on awake suppression and awake reactivation, as those analysis are completely within-subject and thus variability between subjects makes no difference to them. However, the finding that the delay between Days 2 and 4 was related to the learning amount could potentially affect the analyses in Figure 4 where we found that subjects with larger learning showed greater awake suppression. To check if awake suppression may alternatively be driven by the delay between Days 2 and 4, we repeated the analysis from Figure 4 but split our subjects into those who took shorter vs. longer between Days 2 and 4 using a median split (same as in the analysis reported above) and examined whether the two groups differ in the strength of awake suppression. We found that there was no significant difference between the two groups ($T(10)=0.503$, $P=0.633$, unequal variances t-test), suggesting that awake suppression was likely driven by the learning and not by the delay, even though those latter two quantities were related to each other. We have now added these analyses to the Results section of the manuscript and paste the relevant section below:

It should be noted that different subjects experienced different delays between Days 2 and 4 (ranging from three to seven days). We therefore checked whether this variable delay was related to the strength of performance improvement. We divided subjects into those who had a shorter vs. a longer delay between Days 2 and 4 using median split and found that those with longer delays showed greater learning ($T(10)=-2.575$, $P=0.028$). However, the shorter vs. longer delay groups did not differ in the strength of the observed awake suppression ($T(10)=0.503$, $P=0.633$, unequal variances t-test). Therefore, the finding of positive association between awake suppression and learning is unlikely to be driven by the difference in the delay between Days 2 and 4.

10) Line 439, the authors mentioned that 6 of the subjects had already participated in their previous experiment and did not collect their Day 1 scan again. The time interval was from 5 to 15 months (line 447). Could they cite a source or justify a separation of 5 to 15 months not affecting the decoder performance?

We understand the reviewer's concern regarding the long time period between collecting the scans for building the decoder and the scans where the decoder is applied to. However, we note that any drop in the decoder performance should only make it harder for us to find any

significant effects, regardless of whether they are related to awake reactivation or awake suppression. Therefore, any potential drop in performance of the decoder would only work against us. That said, there is also some evidence that decoder performance remains relatively stable even during such long periods of time. For example, several studies recruited patients shortly after a traumatic brain injury (TBI) and followed them for a period of up to 6 months. These studies showed that MVPA decoder performance remained high even though a lot of neural plasticity can be expected for TBI patients (e.g., Schmah et al., 2010). On the other hand, the visual cortex is known to be stable after a critical period in childhood, though some plasticity during adulthood can occur following injury or visual deprivation (Wandell and Smirnakis, 2009; Castaldi et al., 2020). However, our subjects did not have either brain injury or visual deprivation during the intervening 5 to 15 months. We now briefly discuss this issue in the Methods and paste the relevant section below:

The time interval between these six subjects' Day 1 scans from our previous study (Bang et al., 2018) and their new data collection in the current study (Days 2-4) varied from 5 to 15 months. The relatively long period could decrease the decoder performance but such an effect would only make it harder to find evidence for awake suppression or reactivation. Further, the structure and functional properties of the visual cortex are known to be stable during adulthood in the absence of brain injury or visual deprivation (Wandell and Smirnakis, 2009; Castaldi et al., 2020) and substantial MVPA classification performance over periods of many months has been observed even in cases of brain injury (Schmah et al., 2010).

11) Line 447, the authors had mentioned new and old subjects did not show a difference in their behavioral and neural effects. Could the authors create a figure or table showing the results of statistical tests to help the reader?

As requested, we have now created a table that reports the comparison results of awake reactivation, awake suppression, and performance improvement between subjects who previously participated vs. the newly recruited subjects. We also mention these results in the Methods where the distinction between previous vs. new subjects is presented. For the reviewer's convenience, we paste the relevant section from the Methods, as well as the new table (which is added as Supplementary Table 1) below:

Importantly, these six subjects did not differ from the six new subjects in either the observed behavioral or neural effects (**Supplementary Table 1**). Indeed, there was no significant difference between the two groups in the behavioral improvement for either the familiar ($T(10)=-0.914$, $P=0.382$) or novel orientation ($T(10)=0.329$, $P=0.749$), as well as in either the strength of awake reactivation in V1 ($T(7.084)=0.509$, $P=0.626$) or awake suppression in early visual cortex ($T(10)=-0.576$, $P=0.578$).

	Revisited subjects	New subjects	P value
Performance improvement for novel orientation	0.124 ± 0.060	0.090 ± 0.087	0.749
Performance improvement	0.058 ± 0.078	0.179 ± 0.107	0.382

for familiar orientation			
Awake reactivation at post1 in V1	0.533 ± 0.007	0.524 ± 0.016	0.622
Awake suppression at post1 in V1-V3	0.464 ± 0.022	0.486 ± 0.032	0.578

Supplementary Table 1. Behavioral and neural measures for newly recruited and revisited subjects. The two groups did not show any difference in the behavioral and neural effects. The numbers reported for each group are mean ± s.e.m. The P values in the table refer to the results of independent sample t-test between two groups.

12) Line 479, the authors had only one pre-task scan at the beginning of Day 4. Could they justify the reason why they did not have another pre-task scan as a baseline after the anatomical scan?

We agree with the reviewer that this feature of the design should be explained. We did not have another pre-task scan as a baseline after the anatomical scan because we wanted to avoid any possible confounding effect that the preceding exposure could have on the following second pre-task scan. We now added this reasoning in the manuscript and paste the new sentence below:

We did not have separate pre-task scans for each exposure because of the possibility that the pre-task scan for the second exposure could be influenced by the first exposure.

13) Line 486, the authors described a fixation task to prevent conscious rehearsing of a certain orientation. A concern is how an active fixation task affects brain activity in visual areas. Could the authors explain further why the subjects did a visual task but not, for example, an auditory task?

We chose a fixation task instead of a task that involves another sensory modality because we wanted to prevent any conscious rehearsal during post-task scans. We reasoned that under the fixation task, it would be very difficult to rehearse the trained stimuli but such rehearsal may be easier if the task were to use a difficult modality. Furthermore, we wanted to use the same experimental design as our previous paper (Bang et al., 2018), which demonstrated the evidence of awake reactivation during post-training scans. We now include this reasoning in the paper and paste the relevant section below:

During the pre- and post-task scans, subjects performed a fixation task as in our previous work (Bang et al., 2018). We used a visual task (as opposed to a task from a different sensory modality) in order to discourage subjects from consciously rehearsing a certain orientation.

14) Line 547, the authors need to describe in more details how the probability scores are calculated.

We thank the reviewer for pointing out the lack of detail here. We now added more details about how we calculated the probability in the manuscript and pasted the relevant section below:

We used linear sparse logistic regression for decoding (Yamashita et al., 2008). This method implemented by the sparse logistic regression toolbox (SLR toolbox; http://www.cns.atr.jp/~oyamashi/SLR_WEB.html) selects the relevant voxels in the ROIs and calculates their weights for classification.

...

Then we applied the decoder to each 6-volume period of the pre- and post-task scans. The decoder yielded the probability that each 6-volume period was elicited by the familiar or novel orientation. These probability values for each 6-volume period were provided by the output variable named 'Pte' from SLR toolbox. We averaged these probability scores for either the familiar or novel orientation (depending on the analysis) within pre, post1, and post 2 scans. These averaged probability scores served as the probability of classification of the decoder.

15) Line 573, they mentioned using two-tailed parametric tests. However, with a sample size of 12, especially in t-tests for comparing probabilities, it would be helpful to show evidence of normality.

We thank the reviewer for pointing out the necessity of testing the assumptions of the tests that we've used. We confirmed that the assumption of normality was not violated using the Kolmogorov-Smirnov goodness of fit test. In addition, we used Levene's test to assess the equality of variances. When the assumption of equal variances was violated, we used unequal variances tests and reported all such cases. We paste the relevant section from the Methods where we discuss these tests below:

Statistics

For all statistical tests, we used two-tailed parametric tests. We confirmed that the assumption of normality was not violated for all behavioral performance and classification measures by applying the Kolmogorov-Smirnov goodness of fit test. To assess the equality of variances, we used Levene's test. When the assumption of equal variances was violated, we used unequal variances tests and reported all such cases. In addition, we used Mauchly's Sphericity test for all repeated measures ANOVAs to test the assumption of sphericity. When the sphericity assumption was violated, we used Huynh-Feldt correction. We reported all such violations.

16) Line 578, they described the performance improvement. However, the definition is a bit concerning. They have subtracted the S/N ratio of Day 4 from Day 2 and then divided it by Day 2's S/N ratio. However, for the same absolute change of the S/N ratio, subjects with

higher thresholds on Day 2 will be classified as a low improvement. Could the authors describe further their choice of definition?

We agree with the reviewer that there are potentially several ways of classifying performance improvement. The current method of calculating the performance improvement is based on the idea that the performance improvement is best captured by normalizing the change by the initial performance under the assumption that it is the percent change in threshold rather than absolute difference in the threshold that is critical. Our impression is that this method is the most common way to calculate performance improvement in visual perceptual learning (Beard et al., 1995; Furmanski and Engel, 2000; Rokem and Silver, 2010; Wang et al., 2013; Yu et al., 2016). This also the method we used in all of our previous studies (Bang et al., 2018a; Bang et al., 2018b; Bang et al., 2019b). We now add this reasoning in the paper and paste the relevant section below:

We subtracted the threshold S/N ratio after training (Day 4) from that before training (Day 2) and then divided it by the threshold S/N ratio before training (Day 2) to obtain the performance improvement score. This method for calculating performance improvement is common in the field (Beard et al., 1995; Furmanski and Engel, 2000; Rokem and Silver, 2010; Wang et al., 2013; Yu et al., 2016) and is identical to how we calculated performance improvement in our prior work (Bang et al., 2018a; Bang et al., 2018b; Bang et al., 2019b).

References

- Bang JW, Sasaki Y, Watanabe T, Rahnev D (2018) Feature-Specific Awake Reactivation in Human V1 after Visual Training. *J Neurosci* 38:9648-9657.
- Beard BL, Levi DM, Reich LN (1995) Perceptual learning in parafoveal vision. *Vision Res* 35:1679-1690.
- Carr MF, Jadhav SP, Frank LM (2011) Hippocampal replay in the awake state: a potential substrate for memory consolidation and retrieval. *Nat Neurosci* 14:147-153.
- Castaldi E, Lunghi C, Morrone MC (2020) Neuroplasticity in adult human visual cortex. *Neurosci Biobehav Rev* 112:542-552.
- Cheng S, Frank LM (2008) New experiences enhance coordinated neural activity in the hippocampus. *Neuron* 57:303-313.
- Davidson TJ, Kloosterman F, Wilson MA (2009) Hippocampal replay of extended experience. *Neuron* 63:497-507.
- Diba K, Buzsaki G (2007) Forward and reverse hippocampal place-cell sequences during ripples. *Nat Neurosci* 10:1241-1242.
- Euston DR, Tatsuno M, McNaughton BL (2007) Fast-forward playback of recent memory sequences in prefrontal cortex during sleep. *Science* 318:1147-1150.
- Foster DJ, Wilson MA (2006) Reverse replay of behavioural sequences in hippocampal place cells during the awake state. *Nature* 440:680-683.
- Furmanski CS, Engel SA (2000) Perceptual learning in object recognition: object specificity and size invariance. *Vision Res* 40:473-484.
- Giri B, Miyawaki H, Mizuseki K, Cheng S, Diba K (2019) Hippocampal Reactivation Extends for Several Hours Following Novel Experience. *J Neurosci* 39:866-875.

- Han F, Caporale N, Dan Y (2008) Reverberation of recent visual experience in spontaneous cortical waves. *Neuron* 60:321-327.
- Ji DY, Wilson MA (2007) Coordinated memory replay in the visual cortex and hippocampus during sleep. *Nature Neuroscience* 10:100-107.
- Liu ZX, Grady C, Moscovitch M (2018) The effect of prior knowledge on post-encoding brain connectivity and its relation to subsequent memory. *Neuroimage* 167:211-223.
- McNamara CG, Tejero-Cantero A, Trouche S, Campo-Urriza N, Dupret D (2014) Dopaminergic neurons promote hippocampal reactivation and spatial memory persistence. *Nat Neurosci* 17:1658-1660.
- Rokem A, Silver MA (2010) Cholinergic enhancement augments magnitude and specificity of visual perceptual learning in healthy humans. *Curr Biol* 20:1723-1728.
- Schmah T, Yourganov G, Zemel RS, Hinton GE, Small SL, Strother SC (2010) Comparing classification methods for longitudinal fMRI studies. *Neural Comput* 22:2729-2762.
- van de Ven GM, Trouche S, McNamara CG, Allen K, Dupret D (2016) Hippocampal Offline Reactivation Consolidates Recently Formed Cell Assembly Patterns during Sharp Wave-Ripples. *Neuron* 92:968-974.
- Wandell BA, Smirnakis SM (2009) Plasticity and stability of visual field maps in adult primary visual cortex. *Nat Rev Neurosci* 10:873-884.
- Wang R, Cong LJ, Yu C (2013) The classical TDT perceptual learning is mostly temporal learning. *J Vis* 13.
- Wimber M, Alink A, Charest I, Kriegeskorte N, Anderson MC (2015) Retrieval induces adaptive forgetting of competing memories via cortical pattern suppression. *Nat Neurosci* 18:582-589.
- Yu Q, Zhang P, Qiu J, Fang F (2016) Perceptual Learning of Contrast Detection in the Human Lateral Geniculate Nucleus. *Curr Biol* 26:3176-3182.

REVIEWERS' COMMENTS:

Reviewer #1 (Remarks to the Author):

The authors have thoroughly addressed the concerns raised in the prior round of review. I appreciate the inclusion of additional analyses, especially the graded similarity approach which I think strengthens the primary finding. I just have a few minor points to consider:

1. There is a minor discrepancy in terms of evidence for suppression of the familiar orientation in the classification results (Fig. 3) vs. the graded pattern similarity analysis (Supp Fig. 2). V3 shows the same pattern of significant suppression, but there are inconsistencies across V1 and V2. Yet, this inconsistency is not even mentioned in the main results when these results are mentioned. Please at least highlight this difference - although I realize it is more of a qualitative rather than quantitative difference across the analyses (as all of the effects still go in the same direction), it should still be brought to the readers attention, given that the pattern similarity analysis is a supplemental figure.

2. Given the lack of prior evidence for this kind of post-learning suppression, this is a rather intriguing finding. I wonder whether the authors potentially see any similarity between their finding and evidence for suppression in early visual cortex seen on shorter time scales, in the context of working memory. I am specifically thinking of the paper by Lorenc et al. 2020 (Scientific Reports) which finds evidence of suppression of future irrelevant or discarded items in working memory in visual cortex. If the authors think there may be any relationship between these findings, it may be worth mentioning/discussing this in their manuscript.

Reviewer #2 (Remarks to the Author):

Please see the comments by another reviewer. We did the review together.

Reviewer #3 (Remarks to the Author):

The authors revised the manuscript thoroughly and addressed all my previous concerns. I have three minor suggestions to help readers in the finalized manuscript.

1-) Thank you for separating participants exposure order in figures 2 and 3. If adding a legend on top of the figures would not be a big change, then it would help readers.

2-) Line 182, regarding my comment #4, since V1 is concluded to be important for reactivation, the readers would be interested to see the difference in decoder's classification probability against chance level in novel-first and novel-second groups. It would be easier for readers if the authors could add the significance levels they reported in their response to the text while emphasizing the similarities between groups.

3-)Line 185, instead of "Similar results", "No effect of exposure order" would be easier for the reader to follow the results.

Dear Dr. Montague-Cardoso,

Thank you for giving us the opportunity to revise our manuscript. We found the comments by the reviewers very helpful. We have provided a detailed point-by-point response below.

Sincerely,

Ji Won Bang and Dobromir Rahnev

Reviewer 1

The authors have thoroughly addressed the concerns raised in the prior round of review. I appreciate the inclusion of additional analyses, especially the graded similarity approach which I think strengthens the primary finding. I just have a few minor points to consider:

1. There is a minor discrepancy in terms of evidence for suppression of the familiar orientation in the classification results (Fig. 3) vs. the graded pattern similarity analysis (Supp Fig. 2). V3 shows the same pattern of significant suppression, but there are inconsistencies across V1 and V2. Yet, this inconsistency is not even mentioned in the main results when these results are mentioned. Please at least highlight this difference - although I realize it is more of a qualitative rather than quantitative difference across the analyses (as all of the effects still go in the same direction), it should still be brought to the readers attention, given that the pattern similarity analysis is a supplemental figure.

We agree with the reviewer that there are slight differences in the two sets of results where the statistical tests for individual regions did not match perfectly across the two types of analyses. We now highlight this fact in the Results section and point out that the different patterns of significance should be interpreted with caution given that the ANOVAs found no statistically significant differences between the three areas in both cases. We paste the updated section of the Results below:

An examination of this effect in individual areas showed that the reduction of similarity for the familiar orientation was significant in V3 ($T(11)=2.825$, $P=0.017$, Hedges' $g=0.999$), but not in V1 ($T(11)=2.092$, $P=0.060$, Hedges' $g=0.576$) or V2 ($T(11)=0.852$, $P=0.412$, Hedges' $g=0.264$). This pattern of significance for individual brain areas is slightly different from the pattern of significance in our classification analyses (Figure 3). However, both overall ANOVAs found no statistical differences between the three areas and therefore the differences in the pattern of statistical significance for individual areas should be interpreted with caution.

2. Given the lack of prior evidence for this kind of post-learning suppression, this is a rather intriguing finding. I wonder whether the authors potentially see any similarity between their finding and evidence for suppression in early visual cortex seen on shorter time scales, in the context of working memory. I am specifically thinking of the paper by Lorenc et al. 2020 (Scientific Reports) which finds evidence of suppression of future irrelevant or discarded items in working memory in visual cortex. If the authors think there may be any relationship between these findings, it may be worth mentioning/discussing this in their manuscript.

We thank the reviewer for alerting us to the very relevant paper by Lorenc et al (2020) and agree that it is worth mentioning in the Discussion. It is currently difficult to know whether the similarities between the two effects are deep or superficial. Both Wimber et al. and Lorenc et al showed suppression during the task, whereas the suppression in our paper is in the post-task, offline period. We prefer not to speculate on this issue at the moment and instead simply point out to these relevant findings and highlight the fact that suppression in them occurred during the task period. We have now done so and paste the updated sections of the Discussion below:

A related phenomenon of pattern suppression has previously been observed in hippocampus for competing memories during the retrieval of target memories (Wimber et al., 2015) and in early visual cortex for future-irrelevant features during a working memory task (Lorenc et al., 2020) but suppression has not previously been demonstrated for post-task periods.

...

Previous studies have already shown that suppression processes could be studied using fMRI (Wimber et al., 2015; Lorenc et al., 2020), suggesting that determining whether awake suppression exists beyond early visual areas is a tractable question.

Reviewer 2 & 3

The authors revised the manuscript thoroughly and addressed all my previous concerns. I have three minor suggestions to help readers in the finalized manuscript.

1-) Thank you for separating participants exposure order in figures 2 and 3. If adding a legend on top of the figures would not be a big change, then it would help readers.

We thank the reviewer for the suggestion. We added a legend on Figures 2 and 3, and paste these figures below:

Figure 2. Probability that neural patterns are classified as the novel orientation before and after exposure to the novel orientation.

Figure 3. Probability that neural patterns are classified as the familiar orientation before and after exposure to the familiar orientation.

2-) Line 182, regarding my comment #4, since V1 is concluded to be important for reactivation, the readers would be interested to see the difference in decoder's classification probability against chance level in novel-first and novel-second groups. It would be easier for readers if the authors could add the significance levels they reported in their response to the text while emphasizing the similarities between groups.

We agree with the reviewer that adding statistics in line 182 would be helpful for readers. Now we added the significance levels and paste the relevant section below:

Critically, the decoder's classification probability for V1 during post1 was comparable between those subjects who were exposed to the novel orientation first (mean = 0.521, SE = 0.013; $T(5)=1.624$, $P=0.165$, Cohen's $d=0.663$, one-sample t-test) and those who were exposed to the novel orientation second (mean = 0.536, SE = 0.011; $T(5)=3.427$, $P=0.019$, Cohen's $d=1.399$, one-sample t-test), though the probability was significantly higher than chance level for the latter group only.

3-) Line 185, instead of "Similar results", "No effect of exposure order" would be easier for the reader to follow the results.

We thank the reviewer for the suggestion. Now we changed "Similar results" to "No effect of exposure order" as below:

There was similarly no effect of exposure order when examining the percentage change of the decoder's classification between before (pre) and after the exposure (post1) (V1: $T(10)=-0.624$, $P=0.547$, Hedges' $g=0.332$; V2: $T(10)=-0.367$, $P=0.721$, Hedges' $g=0.196$; V3: $T(10)=0.827$, $P=0.428$, Hedges' $g=0.441$; independent sample t-tests).